# Conditional knockout of RAD51-related genes in *Leishmania major* reveals a critical role for homologous recombination during genome replication

Jeziel D. Damasceno[1]*, João Reis-Cunha[2], Kathryn Crouch[1], Dario Beraldi[1], Craig Lapsley[1], Luiz R. O. Tosi[3], Daniella Bartholomeu[2], Richard McCulloch[1]*

1 The Wellcome Centre for Integrative Parasitology, University of Glasgow, Institute of Infection, Immunity and Inflammation, Sir Graeme Davies Building, 120 University Place, Glasgow, United Kingdom, 2 Laboratório de Imunologia e Genômica de Parasitos, Departamento de Parasitologia, Instituto de Ciências Biológicas, Universidade Federal de Minas Gerais, Belo Horizonte, Minas Gerais, Brasil, 3 Department of Cell and Molecular Biology, Ribeirão Preto Medical School, University of São Paulo; Ribeirão Preto, SP, Brazil

* Jeziel.damasceno@glasgow.ac.uk (JDD); Richard.mcculloch@glasgow.ac.uk (RM)

**Data Availability Statement:** Sequences used in this study have been deposited in the European Nucleotide Archive and can be accessed using the accession number PRJEB35031.

## Abstract

Homologous recombination (HR) has an intimate relationship with genome replication, both during repair of DNA lesions that might prevent DNA synthesis and in tackling stalls to the replication fork. Recent studies led us to ask if HR might have a more central role in replicating the genome of *Leishmania*, a eukaryotic parasite. Conflicting evidence has emerged regarding whether or not HR genes are essential, and genome-wide mapping has provided evidence for an unorthodox organisation of DNA replication initiation sites, termed origins. To answer this question, we have employed a combined CRISPR/Cas9 and DiCre approach to rapidly generate and assess the effect of conditional ablation of RAD51 and three RAD51-related proteins in *Leishmania major*. Using this approach, we demonstrate that loss of any of these HR factors is not immediately lethal but in each case growth slows with time and leads to DNA damage and accumulation of cells with aberrant DNA content. Despite these similarities, we show that only loss of RAD51 or RAD51-3 impairs DNA synthesis and causes elevated levels of genome-wide mutation. Furthermore, we show that these two HR factors act in distinct ways, since ablation of RAD51, but not RAD51-3, has a profound effect on DNA replication, causing loss of initiation at the major origins and increased DNA synthesis at subtelomeres. Our work clarifies questions regarding the importance of HR to survival of *Leishmania* and reveals an unanticipated, central role for RAD51 in the programme of genome replication in a microbial eukaryote.

## Author summary

Homologous recombination plays a key role in genome maintenance during cell division, but loss of factors directing the reaction has not been described as being lethal in any

**Funding:** This work was supported by the BBSRC [BB/N016165/1, BB/R017166/1 to RM], the MRC [MR/S019472/1 to RM] and by the EU [Marie Sklodowska-Curie Individual Fellowship, RECREPEMLE, to JD and RM]. The Wellcome Centre for Integrative Parasitology is supported by core funding from the Wellcome Trust [104111, to RM]. The funders had no role in study design, data collection and analysis, decision to publish, or preparation of the manuscript.

**Competing interests:** The authors have declared that no competing interests exist.

microbe. Here, we have used a genetic strategy to selectively induce loss, singly and doubly, of five genes in *Leishmania* that act in homologous recombination, revealing two things. First, loss of any gene related to RAD51, which catalyses homologous recombination, is not immediately lethal, but leads to increasing growth impairment and genome damage accumulation. Second, loss of RAD51 causes a pronounced change in the programme of *Leishmania* genome replication. Thus, we show that homologous recombination in *Leishmania* can be essential, in part due to an unanticipated role in genome transmission.

## Introduction

Homologous recombination (HR) has critical roles in the genome maintenance of all organisms, mainly through repair of double stranded DNA breaks [1]. HR is a multistep repair process initiated by resection of the ends of double-stranded DNA breaks to generate single stranded DNA overhangs. This processing provides access to a key player in HR: the Rad51 recombinase (RecA in bacteria, RadA in archaea)[2], which catalyses invasion of the single-stranded DNA into intact homologous duplex DNA, allowing template-directed repair of the broken DNA site. During evolution, duplications of the Rad51 gene have resulted in so-called Rad51 paralogues [3], a set of factors that are found in variable numbers in different organisms and whose spectrum of roles remain somewhat undefined, at least in part because they can belong to a number of protein complexes. Nonetheless, Rad51 paralogues have been implicated in directly modulating HR, acting on Rad51 HR intermediates [4–6], and in wider repair activities for cell cycle progression [7, 8]. HR reactions mediated by Rad51 [9–12] and modulated by the Rad51 paralogues [13] are also required for resolving DNA replication impediments, by promoting protection and restart of stalled replication forks during replication stress. An even more intimate association between HR and DNA replication has been described in bacteria and archaea, where RecA [14–17] and RadA [18] can mediate DNA replication when origins (the genome sites where DNA synthesis begins during replication) have been removed.

In addition to its roles in promoting genome stability, HR can drive to genome variation, which can cause diseases [19], as well as being a means for targeted sequence change during growth, such as during mating type switching in yeast [20]. Genome variation due to HR is found widely in trypanosomatid parasites, which are single-celled microbes that cause human and animal diseases worldwide. In *Trypanosoma brucei*, HR factors have been clearly implicated in the directed recombination of Variant Surface Glycoprotein (VSG) genes during host immune evasion by antigenic variation [21], as well as in maintenance of the massive subtelomeric VSG gene archive [22, 23]. In *T. cruzi*, HR has been suggested to be a driver of variability in multigene families [24, 25] and in cell hybridisation [26]. Finally, in *Leishmania*, HR related factors have been implicated in mediating the formation or maintenance of episomes, which appear to form stochastically, arise genome-wide and have been implicated in acquisition of drug resistance [27–32]. Whether the same roles for HR extend to widespread, stochastic formation of aneuploidy is unknown [33], but this other form of genome-wide variation has also been implicated in adaptation of the parasite, such as during life cycle transitions and in response to drug pressure [34–38].

Despite emerging evidence for HR roles in *Leishmania* genome change, is it possible that the reaction has wider and deeper functions in genome maintenance and transmission in the parasite? One reason for asking this question is recent observations suggesting that origin

number and distribution in *Leishmania* is unusual, since one study detected only a single site of DNA replication initiation per chromosome [39], while a later study suggested >5000 sites [40]. These data indicate either a pronounced paucity or huge overabundance of origins relative to all other eukaryotes. Alternatively, since in neither study was DNA replication mapping correlated with binding of replication initiation factors, the disparity between the datasets may be due to a widespread, unconventional route for initiation of DNA synthesis acting alongside a small number of conventional origins, perhaps indicating novel strategies for DNA replication that may link with genome plasticity [41]. A second reason for asking about the importance of HR for genome transmission in *Leishmania* is other work that has led to uncertainty about the importance of HR factors for survival of the parasite. *Leishmania* encodes a highly conserved, canonical Rad51 recombinase [30, 42, 43], as well three Rad51 paralogues, referred to as RAD51-3, RAD51-4 and RAD5-6 [29], a slightly smaller repertoire of non-meiotic Rad51-related proteins than is found in *T. brucei* [44–47]. In *L. donovani* [48], unlike in *L. infantum* [30], it has proved impossible to make RAD51 null mutants. Furthermore, while null mutants of RAD51-4 are viable in *L. infantum*, RAD51-3 has been described as being essential, and RAD51-6 nulls were not recovered in the same experiments [29]. Mutation in Rad51 or its relatives has never been shown to be lethal in any single celled eukaryote, notably including *T. brucei* [49], or in prokaryotes, making these observations in *Leishmania* particularly striking.

In this work we sought to resolve the question marks over essentiality of HR factors in *Leishmania* and to test for roles in DNA replication by using conditional gene knockout (KO), comparing the short- and long-term effects of ablating RAD51 and each of its three RAD51 paralogues. Our data show that loss of each gene is, over time, increasingly detrimental to *Leishmania* fecundity, demonstrating that black and white definitions of essential or non-essential are too limiting for HR genes in the parasite. In addition, we show that the functions provided by the RAD51 paralogues are non-overlapping in *Leishmania*, and we reveal that RAD51 plays an unexpected, central role in genome replication, since in its absence the normal programme of DNA replication is profoundly altered.

## Results

### A combined CRISPR/Cas9 and DiCre approach for assessment of gene function in *L. major*

In order to compare the effects of ablating RAD51 and each of the three known *L. major* RAD51 paralogues, we adopted a rapid approach to generate cell lines for conditional induction of a gene KO. For this, we used a cell line constitutively expressing Cas9 and DiCre (Fig 1A). In this strategy, we first used CRISPR/Cas9-mediated HR to exchange the endogenous copy of the genes by a copy flanked by *loxP* sites. In addition, each construct translationally fused copies of the HA epitope to the C-terminus of the gene's ORF. PCR showed this approach to be very efficient for RAD51 and the three RAD51 paralogues, since selection using only puromycin resulted in all wild type (WT) copies of each gene being replaced by floxed and tagged versions after a single transformation (S1 Fig). Because RAD51 and the RAD51 paralogue mutants may generate similar phenotypes [44, 46], since each contributes to HR [29], we used the same approach to modify the *L. major* gene encoding the orthologue of *T. brucei* PIF6. This factor has not been functionally examined in *Leishmania*, but in *T. brucei* PIF6 is the sole known nuclear Pif1 helicase homologue [50]. Since Pif1 helicases have been implicated in modulating DNA replication passage through barriers and during termination [51, 52], and thus operate in distinct aspects of nuclear genome maintenance compared with Rad51 paralogues, we considered this gene could provide a valuable control for the effects of conditional gene KO of the Rad51-related proteins. A number of attempts failed to replace all

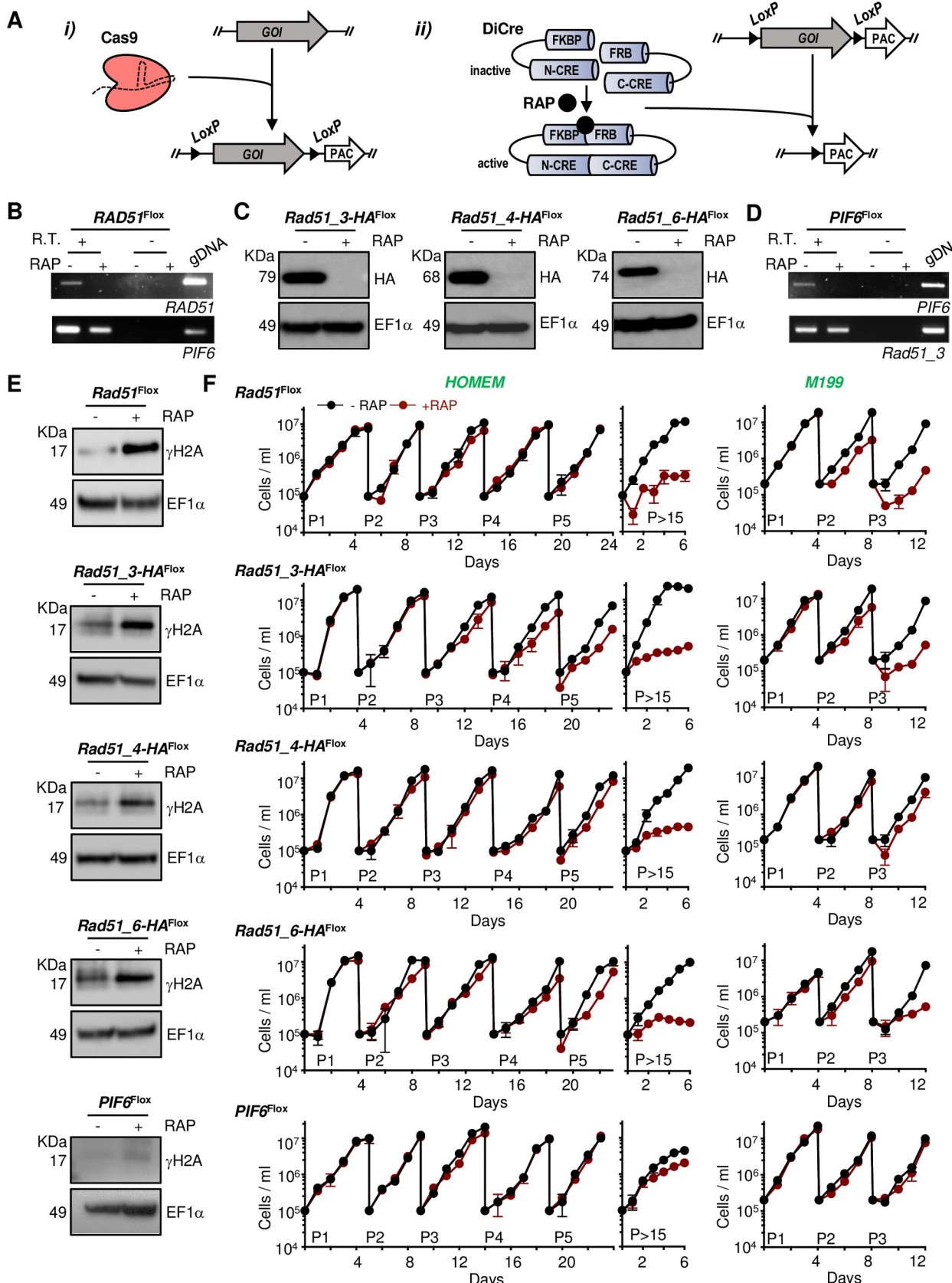

**Fig 1. Combining CRISPR/Cas9 and DiCre allows rapid assessment of homologous recombination gene function by conditional excision. (A)** A cell line was engineered to express both Cas9 and DiCre; i) Cas9 was used to rapidly replace all copies of a gene of interest (GOI) by a version of

the same GOI flanked by *LoxP* sites (GOI^Flox); ii) KO induction was achieved by rapamycin-mediated activation of DiCre, which catalyses excision of GOI^Flox; please refer to S4A Fig for the rapamycin induction strategy. **(B)** and **(D)** RT-PCR analysis of cDNA from the *RAD51*^Flox and *PIF6*^Flox cell lines after 72 h of growth with (+) or without (-) addition of RAP; R.T. (+) and R.T.(-) indicate addition or omission, respectively, of reverse transcriptase in the cDNA synthesis step; amplification of *PIF6* and *RAD51-3* was used as an RT control in B and D, respectively; gDNA indicates a control PCR reaction using genomic DNA as template. **(C)** Western blotting analysis of whole cell extracts of the indicated cell lines after 48 h growth without addition (-RAP) or after addition (+RAP) of rapamycin, leading to DiCre induction; extracts were probed with anti-HA antiserum and anti-EF1α was used as loading control (predicted protein sizes are indicated, kDa). **(E)** Western blotting analysis of whole cell extracts from the indicated cell lines after 48 h growth,–RAP and +RAP; extracts were probed with anti-γH2A antiserum and anti-EF1α was used as loading control (protein sizes are indicted, kDa). **(F)** Growth curves of the indicated cell lines in the presence (+, red) or absence (-, black) of RAP in either HOMEM (left) or M199 (right) medium; cells were seeded at ~$10^5$ cells.ml$^{-1}$ at day 0 and diluted back to that density every 4–5 days for the indicated number of passages (P); growth profile was also evaluated after cells were kept in culture for more than 15 passages (>P15) in HOMEM medium; cell density was assessed every 24 h, and error bars depict standard deviation from three replicate experiments.

WT endogenous *L. major PIF6* gene copies with HA-tagged versions, whereas each copy could be floxed with untagged gene variants (S1 Fig). Growth curves showed that addition of *loxP* sites or the HA tag did not lead to any significant growth impairment for any of the four *RAD51* paralogues or *PIF6* (S2 Fig). However, growth of cells expressing HA-tagged RAD51 was impaired, suggesting the epitope impedes activity of the recombinase (S3 Fig), and so we generated cells with floxed copies of untagged *RAD51* (S1 Fig), which grew normally (S2 Fig) and were used in all subsequent experiments.

Next, KO induction of each gene was attempted by rapamycin-mediated DiCre activation in logarithmically growing cultures of each cell line (Fig 1A, S4 Fig). PCR analysis using DNA from cells after a number of induction rounds ('passages'), where cells were grown from low to high density and repeated by dilution, showed that complete gene excision was achieved after passage 2 for all genes (S4 Fig). Controls without addition of rapamycin showed no gene excision, and unexcised gene copies were undetectable even after >15 passages in the presence of rapamycin (S4 Fig). The rapidity of DiCre mediated loss of the gene products was confirmed by western blotting (*RAD51* paralogues) and RT-PCR (*RAD51*, *PIF6*): signal for all HA-tagged proteins was no longer detectable after 48 h of the second round of KO induction (Fig 1C, S3B Fig), and *RAD51* or *PIF6* cDNA could not be PCR-amplified (Fig 1B and 1D). KO induction of *RAD51* and of each of the *RAD51* paralogues, but not of *PIF6*, resulted in increased levels of γH2A [53] in western blotting analysis (Fig 1E), suggesting accumulation of nuclear DNA damage after loss of any *L.major* RAD51-like protein, but not after ablation of PIF6 (at least during unperturbed growth; see below). To attempt to answer the question of whether or not RAD51 and the RAD51 paralogues are essential in *Leishmania* [29, 48], we measured growth of the parasites for a prolonged period after DiCre-induced gene excision (Fig 1F). At least until passage 4, no growth defect was seen due to loss of RAD51, any of the RAD51 paralogues or PIF6 in HOMEM medium. However, when kept in culture for longer periods (Fig 1F), the *RAD51* KO cells and each of the *RAD51* paralogues KO cells, but not the *PIF6* KO cells, showed marked growth defects, suggesting HR factors that contribute to the catalysis of homology-directed strand exchange might be critical for long-term *Leishmania* viability when cultured in HOMEM. In fact, when grown in M199 medium, growth defects were observed more rapidly after excision of *RAD51* or a *RAD51* paralogue, though again not after excision of PIF6 (Fig 1F). Accordingly, flow cytometry analysis showed that prolonged cultivation after KO induction of *RAD51* and the *RAD51* paralogues, but not *PIF6*, resulted in an increased proportion of cells with less than 1C DNA (S5 Fig), suggesting increased genomic instability, perhaps reflecting the increased levels of γH2A. Taken together, the phenotypes seen after induced loss of the five genes suggest some overlap in functions of RAD51 and its relatives, but a distinct role for PIF6. In addition, the PIF6 data demonstrate that prolonged exposure to rapamycin, or effects of DiCre and Cas9 expression, have a negligible effect on growth in these conditions.

## Loss of RAD51 or RAD51-3, but not RAD51-4 or RAD51-6, impairs DNA synthesis in *L. major*

To ask if the impaired growth seen in four of the five induced KO cells is due to a common defect, we tested the extent of DNA synthesis after each gene deletion. To do this, rapamycin induced and uninduced cells were subjected to a short pulse of IdU labelling followed by immunostaining under denaturing conditions and flow cytometry detection, allowing us to track the level and pattern of DNA synthesis in each cell cycle stage (Fig 2A). Loss of RAD51 or RAD51-3, but not loss of RAD51-4, RAD51-6 or PIF6, resulted in a reduced percentage of IdU-positive cells in the population. Quantification of IdU signal in individual S-phase cells at 48 and 72 h of the second round of KO induction confirmed these effects (Fig 2B): a significant reduction in IdU fluorescence was found at both time points in the rapamycin-induced *RAD51* and *RAD51-3* KO cells compared with their cognate uninduced controls, whereas no such reduction was seen after KO of *RAD51-4*, *RAD51-6* or *PIF6*. These data suggest that only loss of RAD51 or RAD51-3 affects DNA synthesis, meaning the growth impairment seen after loss of RAD51 and its relatives, though similar in extent, might not have a common basis, or loss of DNA synthesis is not the main reason for growth reduction after the induced KO of RAD51 or RAD51-3.

Next, we asked if the observed increases in DNA damage after induced KO of the *RAD51*-like genes all had the same basis by examining γH2A levels across the cell cycle. For this, rapamycin induced and uninduced cells were arrested in G1 using 5 mM hydroxyurea (HU) and then released from arrest by removing HU, sampling at the point of arrest and at various times after release for western blotting (Fig 2C). The patterns of γH2A accumulation revealed notable differences in the cell cycle functions of the five genes (Fig 2C). KO induction of *RAD51* or *RAD51-3* resulted in a pronounced increase of γH2A levels in cells navigating through S-phase up to G2/M, suggesting roles related to the resolution, before cell division, of genome injuries that arise during DNA replication. In contrast, loss of *RAD51-4* or *RAD51-6* did not show any clear evidence for increased γ H2A signal compared with uninduced controls after HU release, suggesting a more limited contribution to tackling replication-associated damage, which seems consistent with the absence of changes in IdU uptake after KO. *PIF6* KO displayed a yet further difference, with increased levels of γH2A only ~6 h after HU release (Fig 2C), when much of the population had passed through S-phase (S6 Fig). These data indicate that loss of PIF6 does in fact result in nuclear damage, but this is more limited than is seen after loss of the RAD51-like genes, and suggests that if the putative helicase has a role in resolving replication problems, this is concentrated in the final steps of DNA replication or even in post-replication steps of the cell cycle.

DNA content in all the samples analysed for γH2A levels was next analysed by flow cytometry (S6 Fig). Intriguingly, no pronounced changes in cell cycle progression after release from HU arrest were observed upon KO induction of any of the HR genes or *PIF6* (S6 Fig), perhaps suggesting cell cycle checkpoints are not enacted by the gene KOs, despite clear DNA damage accumulation after loss of RAD51 or its paralogues. Altogether, these data suggest that *Leishmania* RAD51 and RAD51-3, specifically amongst the genes examined, have roles in promoting effective DNA synthesis and their absence results in increased nuclear genome damage during S-phase.

## Ablation of RAD51 or RAD51-3, but not RAD51-4, results in widespread mutagenesis

We next sought to determine if loss of the HR genes results in genome instability by using short-read Illumina sequencing of DNA from *RAD51*, *RAD51-3* and *RAD51-4* KO cells after

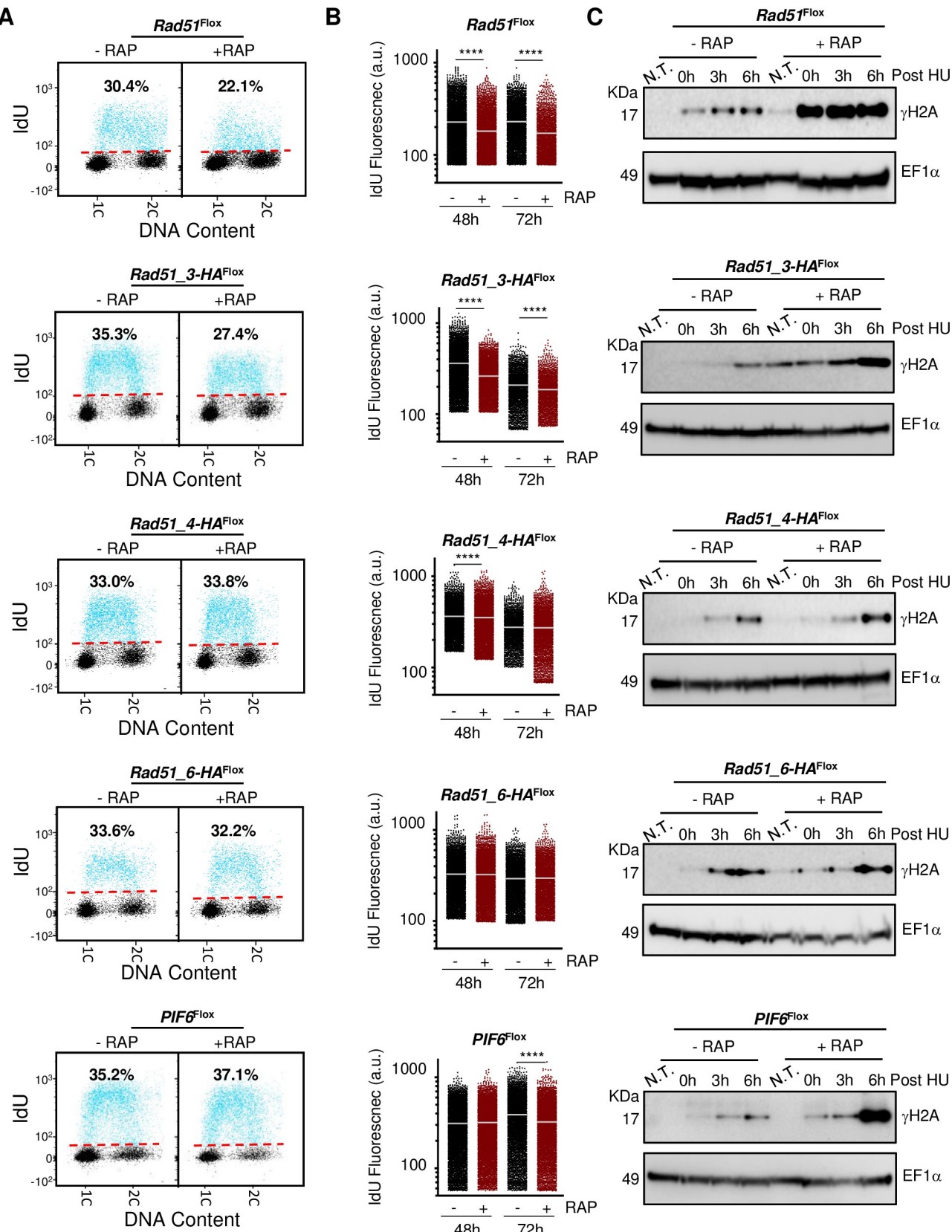

**Fig 2. Analysis of DNA synthesis and cell cycle-dependent accumulation of DNA damage upon induced knockout of homologous recombination factors.** **(A)** Representative dot plots from flow cytometry analysis to detect DNA synthesis in the indicated cell lines; after 48 h growth without (-RAP) or with (+RAP) addition of rapamycin, inducing DiCre, cells were incubated with IdU for 30 min and IdU fluorescence was detected under denaturing conditions; 30,000 cells were analysed per sample; 1C and 2C indicate single DNA content (G1) and double DNA content (G2/M), respectively; dashed red lines indicate the threshold used to discriminate negative (black dots) from IdU-positive (blue dots) events; inset numbers indicate total percentage of IdU-positive events relative to the whole population. **(B)** Quantitative analysis of IdU fluorescence following gene excision in the indicated cell lines; after the indicated times of -RAP or +RAP growth, cells were labelled as in (A); fluorescence from IdU positive S phase cells is plotted as arbitrary units (a.u.); at least 2,000 cells were analysed in each time point and horizontal white lines indicate the mean; differences between -RAP or +RAP cells were tested with a Kruskal–Wallis test (**** denotes P< 0.0001). **(C)** Western blot analysis of whole cell extracts; 48 h after rapamycin DiCre induction (+RAP), or in controls cells with induction (-RAP), the indicated cell lines were left untreated (N.T.) or were treated with addition of 5 mM hydroxyurea (HU) for 8 hrs; cells were collected at the indicated time points after HU removal; extracts were probed with anti-γH2A antiserum and anti-EF1α was used as loading control (predicted protein sizes are shown, kDa).

growth for two and six passages in the presence of rapamycin, as well as in the same cells grown without rapamycin (Fig 3A). In each case, mapping of reads to the genome showed specific loss of sequence around the *loxP*-flanked gene, confirming KO induction (Fig 3B). To understand the effects of HR factor loss, we measured the number of single nucleotide polymorphisms (SNPs) in the induced and uninduced cells after passage two and six (P2 and P6) by comparing these genome sequences to the reference genome; in addition, to clearly determine the effect of DiCre excision, SNPs that were common to the two time points in the induced and uninduced cells were discarded (Fig 3C). Irrespective of whether or not gene loss was induced, SNPs accumulated during growth of *L. major* and the consequences of loss of the three HR factors was not equivalent (Fig 3D). Loss of *RAD51* reduced the level of SNPs relative to uninduced cells at P2, while a significantly increased accumulation of SNPs was seen by P6. In contrast, loss of *RAD51-3* caused a small but significant increase of SNPs at P2, which was no longer detectable at P6. Finally, there was no evidence that loss of *RAD51-4* increased SNP accumulation relative to uninduced cells at either passage. To ask if this pattern of mutagenesis is only seen with SNPs, we also measured insertions and deletions (InDels) in the same cells (S7 Fig), with the same differential patterns seen in the three different mutants (S7C Fig).

Strikingly, when new SNP or InDel density was plotted individually for each of the 36 chromosomes, it became apparent that the increase in SNPs with or without gene KO was not random across the genome, as the smaller chromosomes tended to present a higher density of new SNPs and InDels than the larger chromosomes (Fig 3E, S7D Fig). To ask further if there is localised accumulation of mutations in the *L. major* genome, we examined SNP density proximal to the 'strand switch regions' (SSRs) within each chromosome where multigene transcription initiation and/or termination occurs (Fig 3F), and a subset of which are where MFA-seq has mapped DNA replication initiation (i.e. are predicted replication origins)[39]. This mapping revealed that SNP density peaks around the SSRs, suggesting these loci are hotspots for mutation. Furthermore, SSR proximity mapping of SNPs confirmed a difference between the three gene KOs. Following loss of *RAD51* a decrease in SNP accumulation was seen relative to uninduced controls, and this was accounted for by reduced levels pf SNPs at origin-active SSRS, with little evidence of a change at origin-inactive SSRs. The opposite effect was observed after *RAD51-3* KO, with increased SNP levels at origin active SSRs, and no change in SSR-proximal SNP density was found after *RAD51-4* KO. Thus, the two HR factors whose loss was found to affect global DNA synthesis were also seen to affect SNP accumulation around SSRs, unlike *RAD51-4* KO, which did not show an effect on DNA synthesis. A clear peak of new InDels in the uninduced cells was less apparent than was seen for new SNPs (compare S7E Fig with Fig 3F). However, InDels appeared to accumulate to a greater extent around origin-active than origin-inactive SSRs upon *RAD51* KO (S7E Fig), while such effects were less clearly seen upon *RAD51-3* KO, and no difference was seen in *RAD51-4* KO cells compared with controls. In total, therefore, InDel accumulation proximal to SSRs upon HR KO was more modest than

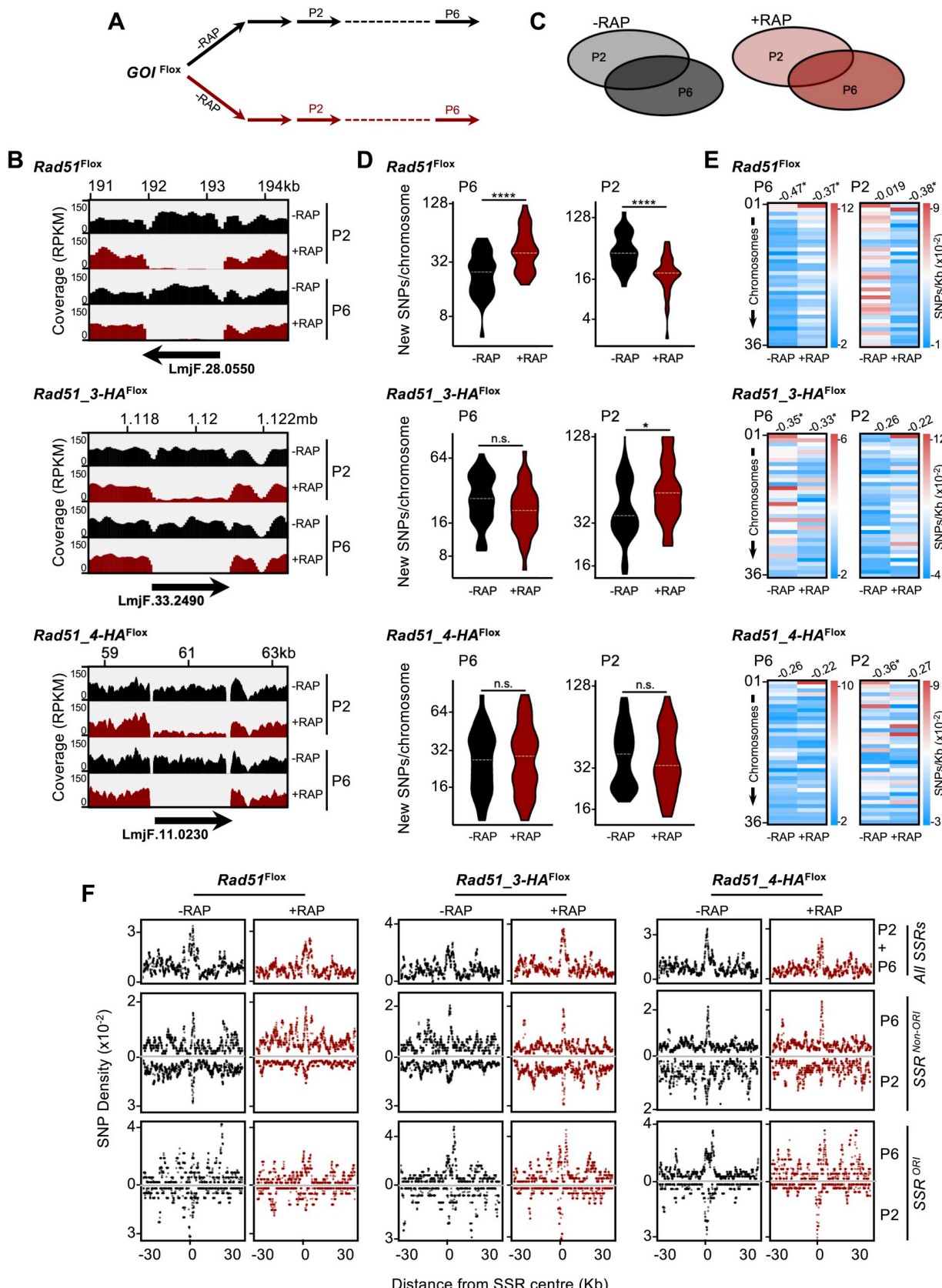

**Fig 3. Whole genome sequencing reveals mutagenesis upon induction of homologous recombination factor gene knockouts. (A)** $GOI^{Flox}$ cell clines were grown in the absence (-) or presence (+) of Rapamycin (RAP); genomic DNA was extracted at passages (P) 2 and 6 and subjected to deep sequencing. **(B)** Sequence read depth around the targeted gene loci in the indicated cell lines in rapamycin-induced (+RAP) and uninduced (-RAP) cells at the indicated times; coverage tracks were generated with deepTools, using the bamCoverage tool [92] and ignoring duplicated reads; RPKM normalization was used to allow comparison across samples. **(C)** SNPs relative to the reference genome were identified; events common to P2 and P6 were discarded; events exclusively found in P2 or P6 were considered for the following analysis. **(D)** Quantification of the number of SNPs detected in P2 and P6; data are represented as violin plots, where shape indicates the distribution of pooled data and horizontal dotted white lines indicate the median; differences were tested with a Mann-Whitney test; * P<0.05, and ****P<0.0001. **(E)** Heatmaps representing density of new SNPs (SNPs/Kb) detected in the indicated passages; numbers at the top of each row indicate Pearson correlation between SNPs density and chromosome size; where correlation is significant, it is indicated by * P<0.05, **P<0.005 and ***P<0.001. **(F)** Metaplots of normalized SNP density (SNPs/Kb) in P2 and P6 is shown for 60 kb of every chromosome centred around the MFAseq-mapped replication origins ($SSR^{ORI}$, n = 36), and for 60 kb around all other strand switch regions that did not show origin activity ($SSR^{non-ORI}$, n = 95).

SNPs, but again any distinct accumulation around SSRs was most clearly detected after loss of RAD51 or RAD51-3, the two factors our data implicate in global DNA synthesis. Moreover, loss of RAD51 appears to have a distinct effect on SNP and InDel accumulation, which is primarily seen at predicted origin-active SSRs.

Given that excision of *RAD51* and *RAD51-3* altered the frequency of SNP accumulation, we next examined the types of mutation that resulted in the SNPs (S8 Fig). Though the data showed some bias towards specific forms of base substitution, it was not clear that induced loss of any of the HR genes altered this pattern relative to uninduced controls and nor was any difference seen when comparing loss of RAD51, RAD51-3 or RAD51-4.

To examine the effects of loss of either RAD51 or RAD51-3 further, we examined survival of the KO cells relative to uninduced controls in the presence of increasing concentrations of phleomycin and camptothecin, both of which cause DNA double-strand breaks, and HU, which inhibits ribonucleotide reductase and impairs DNA synthesis (S9 Fig). As expected for predicted DNA double-strand break repair factors, KO induction of either *RAD51* or *RAD51-3* lead to increased sensitivity to phleomycin and camptothecin. However, only *RAD51-3* KO led to increased sensitivity to HU. Consistent with this growth difference, levels of SNP accumulation after exposure to and release from HU treatment (S10A and S10B Fig) also differed between the two KOs. Both genome-wide (S10C Fig) and in each chromosome (S10D Fig), fewer HU-induced new SNPs were detected in *RAD51* KO cells exposed to HU compared with uninduced, whereas KO of *RAD51-3* did not have such an effect after HU exposure. Despite this differential effect, HU treatment did not alter the SNP mutation profile, with or without gene excision (S11 Fig: compare to S8 Fig). Moreover, unlike for SNPs, loss of either HR gene caused a significant increase in InDel levels, with or without exposure to HU (S10C and S10D Fig), and HU-induced replication stress did not clearly change the SNP or InDel patterns around SSRs upon *RAD51 or RAD51-3* KO (S10E Fig) relative to the changes seen in the absence of HU (Fig 3, S7 Fig). Altogether, these data reinforce the view that, despite loss of RAD51 or RAD51-3 both causing impaired DNA synthesis, the roles of the two factors in maintaining the *L. major* genome differ.

## Generation of conditional double gene mutants by CRISPR/Cas9 and DiCre in *Leishmania*

To date, the analysis of single gene conditional KOs has implicated only RAD51 and RAD51-3 amongst the four HR factors in DNA synthesis. One explanation may be that RAD51-4 and RAD51-6 act redundantly in *Leishmania*. To test this, we used the combined CRISPR/Cas9, DiCre approach to attempt to make conditional double gene KOs of the Rad51-paralogues in all possible combinations (Fig 4A). First, CRISPR/Cas9 was used to generate floxed copies of either *RAD51-3* or *RAD51-4* using gene-specific puromycin-resistance constructs (described

above). Next, the resulting cell lines were subjected to a second round of CRISPR/Cas9 engineering to delete the endogenous copies of *RAD51-4* or *RAD51-6* in the *RAD51-3-HA^flox^* cells, or *RAD51-6* in the *RAD51-4-HA^flox^* cells. PCR on G418-resistant clones for each of the three cell lines (S12 Fig) showed that it was possible to delete all copies of either *RAD51-4* or *RAD51-6*, retaining floxed copies of *RAD51-3-HA* or *RAD51-4-HA*. DiCre-mediated KO of the floxed gene was then induced by addition of rapamycin. PCR analysis (S13 Fig) showed complete excision of *RAD51-3-HA* or *RAD51-4-HA* after the second round of rapamycin-mediated DiCre induction in all three cell lines, thereby generating cells devoid of two *L. major* Rad51-paralogues genes simultaneously (*RAD51-4* and *RAD51-3*; *RAD51-6* and *RAD51-3*; or *RAD51-6* and *RAD51-4)*. Complete loss of the genes was maintained after cells were grown for more than 15 passages (S13 Fig).

Western blotting (Fig 4B) showed that HA-tagged RAD51-3 or RAD51-4 protein in the null mutants was undetectable after rapamycin induction of gene KO. The *RAD51-4* and *RAD51-6* null mutants, prior to induction of conditional excision of the second gene, showed increased levels of γH2A compared with the background cell line (Fig 4C). DiCre-mediated gene excision, leading to the three different double gene KOs, did not result in further increases in the levels of the phosphorylated histone. The lack of further impairment when two genes are lost relative to one was broadly consistent with growth curves (S14A Fig) and with flow cytometry analysis of DNA content (S14B Fig): conditional excision of the second gene did not lead to worsening of growth or to increased numbers of aberrant cells with less than 1C content compared with uninduced single gene null cells. Intriguingly, the *RAD51-4* and *RAD51-6* null mutants, even without induction of the second gene KO and having been selected as clones, appeared to recapitulate the long-term growth impairment seen upon prolonged cultivation after DiCre excision of a single HR gene (compare S14A Fig with Fig 1F).

IdU labelling followed by FACS analysis of each of the three cell lines (Fig 4D), before and after rapamycin induction, did not show a change in the proportion of cells that incorporated the nucleotide analogue. Quantification of IdU fluorescence levels of individual S-phase cells (Fig 4E) did not reveal a consistent pattern of fluorescence decrease upon simultaneous KO of two *RAD51* paralogs. However, a signal reduction was detected 72 h after *RAD51-3* KO induction in both the *RAD51-4* and *RAD51-6* null backgrounds, consistent with loss of only RAD51-3 amongst the paralogues impeding DNA synthesis (Fig 2A and 2B). In contrast, though a mild loss of fluorescence was seen at 48 h when KO of *RAD51-4* was induced in the *RAD51-6* null background, this was not seen at 72 h (Fig 4E). These data reinforce the suggestion that RAD51-3 alone among the three RAD51 paralogues plays a role in *L. major* DNA replication, and indicate that RAD51-4 and RAD51-6 do not obviously act redundantly in such a function.

To ask if the induction of a double *RAD51* paralogue gene KO had similar effects on genome instability relative to what was seen after induction of single gene KOs (Fig 3), we again performed Illumina sequencing of DNA from the *RAD51-4* null *RAD51-3-HA^flox^* cells after two and six passages of growth with and without rapamycin (Fig 5A and 5C). Read mapping to the genome showed the *RAD51-4* gene was absent and that DiCre induction lead to removal of *RAD51-3* (Fig 5B). Like what was seen in the *RAD51-3* KO (Fig 3D), SNPs accumulated to a greater extent across the genome (Fig 5C) and in each chromosome (Fig 5D) in the *RAD51-3 RAD51-4* double KO cells at P2, but no difference was seen at P6. Furthermore, increased SNP accumulation around SSRs upon induction of double a *RAD51-4 RAD51-3* KO seemed comparable with the RAD51-3 KO, and distinct from *RAD51-4* KO cells (Fig 5E). Finally, broadly comparable changes in levels and patterns of InDel accumulation were seen after induction of the double *RAD51-4 RAD51-3* KO (S15A–S15C Fig) and *RAD51-3* single KO (S7 Fig). Thus, these data, allied to previous DNA synthesis and cell cycle-dependent

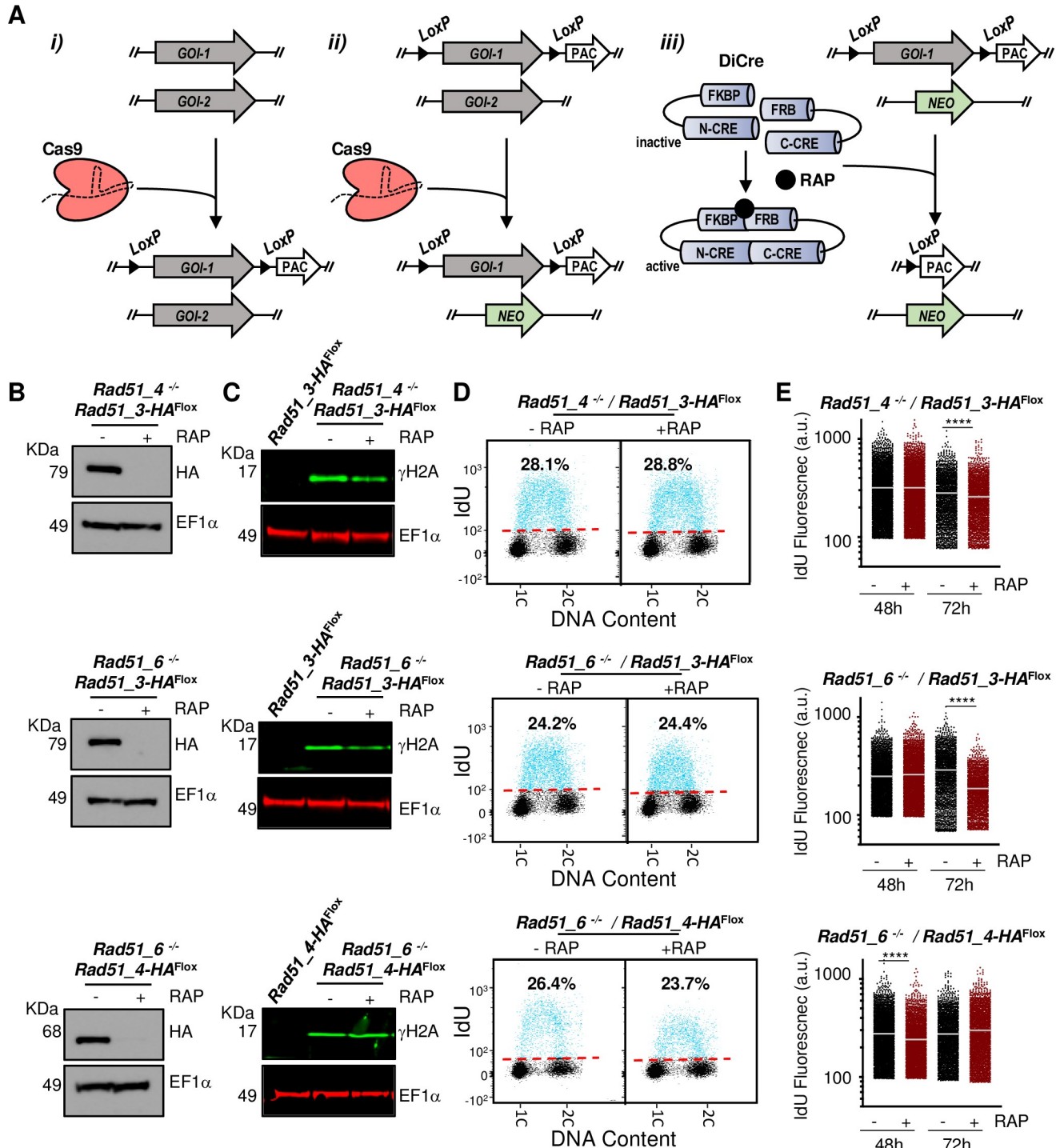

**Fig 4. Combining CRISPR/Cas9 and DiCre to generate double null mutant cells. (A)** In the cell line expressing both Cas9 and DiCre, the following approach was used: i) Cas9 was used to replace all copies of the first GOI (GOI-1) by GOI-1^Flox; ii) Cas9 was used to replace all copies of the second GOI (GOI-2) by a neomycin resistance cassette (NEO), making this gene a null mutant (-/- panel B); iii) KO induction of GOI-1^Flox is achieved by rapamycin-mediated activation of DiCre, generating a double null mutant cell line for both GOIs. **(B)** Western blotting analysis of whole cell extracts after 48 h growth without addition of rapamycin (-RAP) or after addition of rapamycin (+RAP), leading to DiCre induction; extracts were probed with anti-HA antiserum and anti-EF1α was used as loading control (predicted protein sizes are indicted, kDa). **(C)** Western blotting analysis of whole cell extracts from the indicated cell lines after 48 h of–RAP and +RAP growth; extracts were probed with anti-γH2A antiserum (green) and anti-EF1α (red) was used as loading control. **(D)** Representative dot plots from flow cytometry analysis to detect DNA synthesis in the indicated cell lines; after 48 h of–RAP or +RAP growth, cells were incubated with IdU for 30 min and IdU fluorescence was detected under denaturing conditions; 30,000 cells were analysed per sample; 1C and

2C indicate single DNA content (G1) and double DNA content (G2/M), respectively; dashed red lines indicate the threshold used to discriminate negative (black dots) from IdU-positive (blue dots) events, and inset numbers indicate total percentage of IdU-positive events relative to the whole population **(E)** Quantitative analysis of IdU fluorescence following gene excision in the indicated cell lines; after the indicated times of -RAP or +RAP growth, cells were labelled as in (D); fluorescence from IdU positive cells is plotted as arbitrary units (a.u.); at least 2,000 cells were analysed in each time point; horizontal white lines indicate the mean; differences were tested with a Kruskal–Wallis test (**** denotes P< 0.0001).

damage analysis, suggest a complex mixture of common and separate functions for these two RAD51 paralogues, probably related to epistatic interactions among them.

## Loss of RAD51 leads to changes in the DNA replication landscape of *Leishmania*

Our analyses until this point indicated impairment of DNA synthesis after loss of either RAD51 or RAD51-3, including evidence for distinct effects of the gene KOs. However, if and

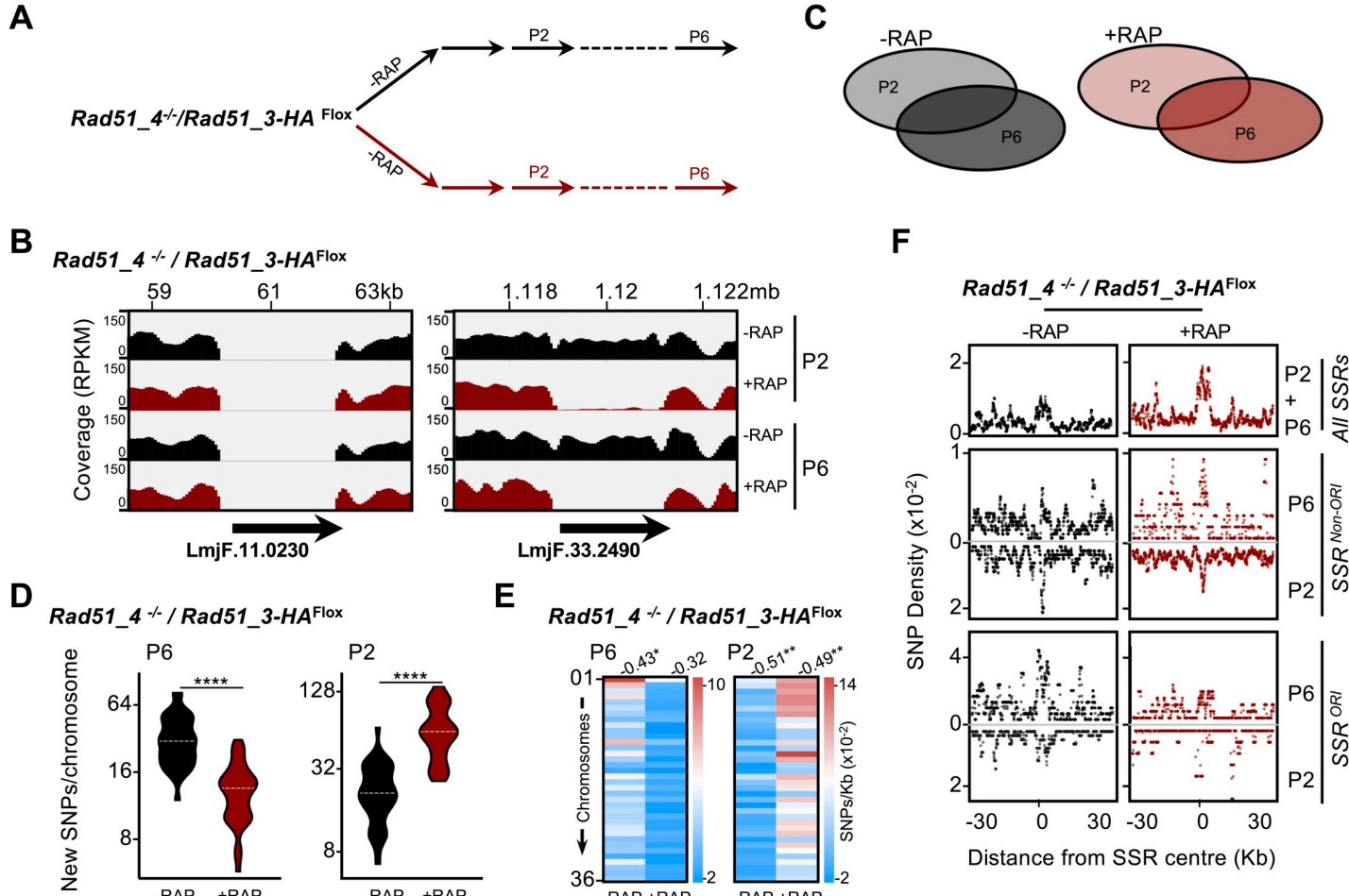

**Fig 5. Mutagenesis upon induction of single or double mutants of two RAD51 paralogues. A)** A *RAD51-4$^{-/-}$/ RAD51-3-HA$^{Flox}$* cell line was grown in the absence (-) or presence (+) of Rapamycin (RAP); genomic DNA was extracted at passage (P) 2 and 6 and subjected to deep sequencing. **(B)** Sequence read depth around the targeted gene loci in +RAP and -RAP cells at the indicated times; coverage tracks were generated with deepTools, using the bamCoverage tool [92] and ignoring duplicated reads; RPKM normalization was used to allow comparison across samples. **(C)** SNPs relative to the reference genome were identified; events common to P2 and P6 were discarded; events exclusively found in P2 or P6 were considered for the following analysis. **(D)** Quantification of the number of SNPs detected at P2 and P6; data are represented as violin plots, where shape indicates the distribution of pooled data and horizontal dotted white lines indicate the median; differences were tested with Mann-Whitney test; ****P<0.0001. **(E)** Heatmap representing density of SNPs (SNPs/Kb) detected in the indicated passages; numbers at the top of each row indicate Pearson correlation between SNPs density and chromosome size; when correlation is significant, it is indicated by * P<0.05 and **P<0.005. **(F)** Metaplots of normalized SNP density (SNPs/Kb) at P2 and P6 is plotted +/- 30 Kb around the centre of either *SSR$^{ORI}$* (*n* = 36) or *SSR$^{non-ORI}$* (*n* = 95) for the indicated cell lines.

how the two factors might contribute to the programme of DNA replication in *L. major* was not known. To address this question, we performed MFA-seq analysis, comparing read depth across the chromosomes in DNA extracted from replicating and non-replicating cells, thereby identifying sites of replication initiation and patterns of fork progression [39, 54, 55]. To do this, we used a modified version of MFA-seq compared to what we have described previously: rather than using FACS to isolate S- and G2-phase cells, we isolated DNA from *L. major* cells in logarithmic (log) or in stationary phase growth, sequenced each using Illumina technology, and mapped the ratio of reads in the two populations. Fig 6A shows these data as Z-scores across two chromosomes, where positive signal represents regions where the mean read depth in the log phase cells is greater than the mean of the stationary cells (S16 Fig shows further chromosomes). This modified MFA-seq approach will be described in detail elsewhere (BioR-Xiv 10.1101/799429), but two primary features of *Leishmania* replication are revealed relative to MFA-seq based on S/G2 phase read depth ratios using cell cycle-sorted cells [39]. First, we confirm the predominant use of single origins in each chromosome, most of which are centrally disposed in the molecules and each coinciding with SSRs. Second, we now detect weaker sites of replication initiation that were not seen previously and are proximal to one or both telomeres in the chromosomes (see -RAP data in Fig 6A, which are highly comparable with WT cells; BioRXiv 10.1101/799429).

To ask about the effect of RAD51 or RAD51-3 loss on DNA replication, we performed MFA-seq in cells induced for KO by growth in rapamycin (48 h, second round of induction), as well as in control cells without rapamycin grown for the same time. Comparing induced *RAD51* KO cells with uninduced cells revealed a striking variation in MFA-seq pattern: loss of MFA signal was seen at the main origin that had previously been mapped within each chromosome [39], while an increase in MFA-seq signal was seen at the extremities of the chromosomes (Fig 6A, S16 Fig). In the induced *RAD51-3* KO cells, the same differential effect was less obvious: though there was potentially some loss of MFA-seq signal at the main origin, increase in subtelomeric signal was not apparent. To examine this genome-wide, we generated meta-plots of the MFA-seq signal in the different cells (Fig 6B and 6C). Profiling of MFA signal around the main origin (Fig 6B) revealed considerable consistency in amplitude and width of the peaks, both for the 36 origins within one cell and between the two uninduced cells, confirming there is little variation in timing or efficiency of DNA synthesis initiation at all main origins in the WT *L. major* population [39]. The same profiling confirmed that loss of MFA-seq signal around the main origins was a genome-wide effect upon *RAD51* KO, with lowered amplitude and width, and was less pronounced upon KO of *RAD51-3*. These data indicate that loss of RAD51 leads to a more pronounced decrease in replication initiation activity at the main origins compared with RAD51-3 loss, which appears consistent with decreased SNP accumulation at SSRs after *RAD51* KO but not after *RAD51-3* KO (Fig 3F). Profiling of the MFA-seq mapping at the chromosome ends (Fig 6C) showed that loss of RAD51, but not RAD51-3, resulted in a gain of signal in the subtelomeres, which again was strikingly consistent in amplitude and width across all chromosomes. Altogether, these data indicate that loss of DNA replication initiation at the main, chromosome-central origins due to ablation of RAD51 is accompanied by increased DNA replication in the subtelomeres, a shift in the replication programme that is not clearly seen after loss of RAD51-3. In addition, these observations further confirm that RAD51 and RAD51-3 play distinct roles in *Leishmania* DNA replication.

It remains possibile that the above effects could be due to replication stress and DNA damage accumulation upon *RAD51* KO (Fig 1E). Though this possibility appears to be argued against by the effects of *RAD51-3* KO, which also resulted in increased levels of DNA damage but did not lead to an alteration in the MFA-seq profile, we attempted to test this idea further

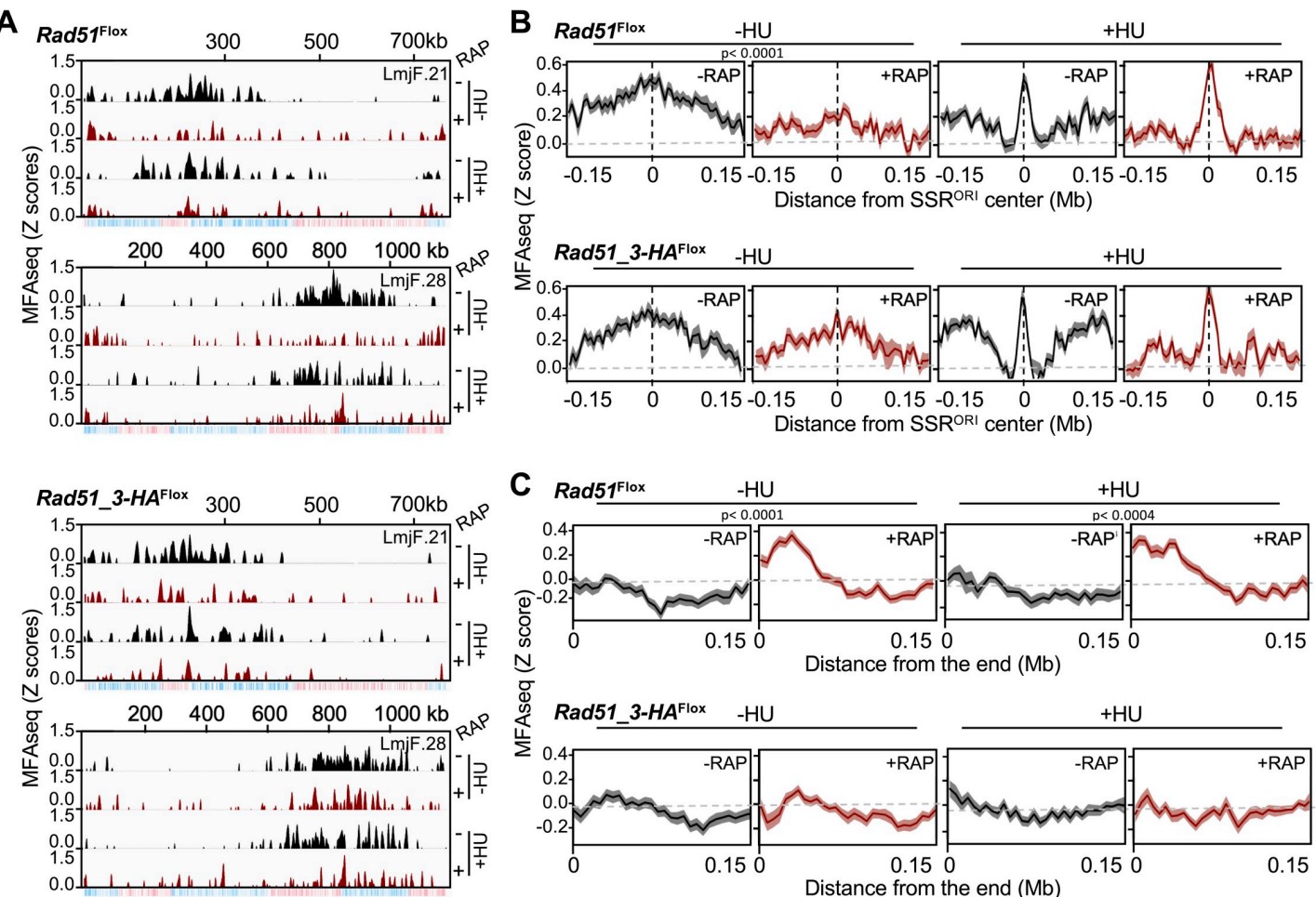

**Fig 6. Genome-wide mapping reveals impaired initiation of DNA replication upon induced knockout of RAD51. (A)** Graphs show the distribution of sites of DNA replication initiation across two complete chromosomes in the indicated cell lines, in each case grown in the absence (-RAP) or the presence (+RAP) of rapamycin; MFA-seq signal after cells were incubated with 5mM HU for 8 hours is also indicated. MFA-seq signal is represented by Z-scores across the chromosomes, calculated by comparing read depth coverage of DNA from exponentially growing cells relative to stationary cells; the bottom track for each chromosome displays coding sequences, with genes transcribed from right to left in red, and from left to right in blue. **(B)** Metaplots of MFA-seq signal found in every chromosome, centred on the previously mapped constitutive DNA replication origin (SSR$^{ORI}$) ±0.15 Mb, in -RAP and +RAP cells, and in the absence (-) or presence of HU (+). **(C)** Metaplots of MFA-seq signal across 0.15 Mb of sequence from all chromosomes ends in -RAP and +RAP cells, and with (+HU) or without (-HU) growth in the presence of HU. In B and C, *p* values were determined using Wilcoxon test by comparing -RAP with +RAP cells within each -HU and +HU pair.

by searching for changes in MFA-seq profile upon treatment with 5 mM HU for 8 hours, which also causes replication stress associated with DNA damage levels (Fig 2C). HU treatment resulted in a narrower MFA-seq peak at the centre of the origin-active SSRs, surrounded by two shoulders (Fig 6B). This effect may be explained by impairment of bi-directional replication fork movement around the origins, since the cell cycle is mainly arrested at the G1/S transition by this treatment (S6 Fig). Only subtle changes in this HU-induced MFA-seq signal were observed after loss of RAD51 or RAD51-3, consistent with the FACS analysis showing no evidence for defective cell cycle arrest upon absence of the HR factors (S6 Fig). These data may indicate that the effect of RAD51 loss on DNA replication around origins in untreated cells is not due to roles at the early stages of S phase, but rather later in the cell cycle. MFA-seq signal near the chromosome ends upon RAD51 KO was very comparable between untreated and HU-treated cells, suggesting these replication profile changes are mainly concentrated at the

early stages of S phase. Overall, HU treatment suggested the pronounced change in replication profile upon RAD51 loss cannot be simply be accounted for replication stress or DNA damage accumulation, suggesting RAD51 is more intimately involved in the *L. major* DNA replication programme.

## Discussion

Homologous recombination is known to be important for episome formation in *Leishmania* [29–31], but the depth and breadth of how the process contributes to genome plasticity and transmission has not been fully explored. Here, we sought to answer two main questions. First, are RAD51 and RAD51 paralogues essential in *Leishmania*, given the conflicting data on the ability to generate and propagate null mutants in different species? Second, given the potentially novel distribution of mapped origins in the *Leishmania* genome, does HR play a central role in *Leishmania* genome duplication? Using a rapid, conditional knockout approach that combines CRISPR/Cas9 gene modification and DiCre gene excision, we show that loss of any RAD51-like gene is not immediately lethal to promastigote *L. major* but is increasingly detrimental over time, with phenotype penetrance dependent on cultivation conditions. Thus, we suggest that binary definitions of essential or non-essential for HR-related genes inadequately describe their contribution to parasite biology. In addition, our data reveal that loss of RAD51 and RAD51-3, uniquely among the four HR proteins examined, impairs DNA replication, though their roles are distinct, with RAD51 playing an unexpected and central role in DNA replication initiation.

Null mutants of *RAD51* and *RAD51-4* have been described in *L. infantum* [29, 30], whereas *RAD51* mutants could not be recovered by CRISPR/Cas9 gene targeting in *L. donovani* [48]. In addition, *RAD51-3* and *RAD51-6* null mutants were not recovered by two-step homology-directed gene deletion in *L. infantum* [29]. The conditional KO approach used here aids understanding of this complex biology, since we show that excision of any of these HR factors has no immediate impact of parasite fitness, but rather causes a progressive slowing of growth, presumably due to accrual of problems. Nonetheless, it is clear that *Leishmania* null mutants of *RAD51* (in *L. infantum*)[30], and the RAD51 paralogues *RAD51-4* and *RAD51-6* (this work, in *L. major*), can be generated. Similarly, null mutants of each *RAD51*-related gene have been described in *T. brucei* [44, 46]. It is possible such variation in importance reflects species-specific aspects of HR function. However, it is also conceivable that conventional, two step transformation approaches to generate null mutant clones allows for selection of cells possessing compensatory changes that lead to survival—an adaptation that may not always be recovered when using CRISPR-Cas9 to simultaneously ablate both alleles [48], or that emerged in the timeframe we have used during conditional gene excision (this work). What such adaptations might be is unclear, but the abundant mutations we describe after loss of HR genes may provide a genetic basis for their generation. In addition, RAD51-directed HR is not the sole route for repair of DSBs in any trypanosomatid [56–62], though whether such pathways can increase in activity in the absence of RAD51-directed HR has not been tested. What aspects of *Leishmania* genome function degrade with time after loss of RAD51 and its relatives remain to be fully characterised, though our data suggest these effects may differ for the different RAD51 paralogues, and for RAD51.

Conditional loss of RAD51 and RAD51-3 led to genome-wide increases in SNPs, as well as increased amounts of in InDels. Loss of RAD51-4 did not lead to such obvious mutation levels and the effect of RAD51-6 loss remains to be determined. Nonetheless, these data demonstrate a widespread role for HR related proteins in *Leishmania* genome stability. In addition, we did not attempt to test for larger genome changes, but translocations have been described in

MRE11 [27], which can guide RAD51 function. Our mapping of SNPs also revealed two unusual features of the patterns of mutagenesis in *L. major*. First, there was a notable chromosome size-dependence on levels of SNP accumulation, with smaller chromosomes tending to have higher SNP densities when compared with the larger chromosomes. This was common to each conditional gene KO, indicating it is general feature of *L. major* chromosome biology. However, the basis of this effect is unclear. Might it relate to differences in gene expression or nucleosome occupancy? No data that we are aware of supports such a suggestion, and the commonality of multigene transcription in all chromosomes appears to argue against it [63]. If not gene expression, then the chromosome size-dependence of SNP density might reflect the limitations of predominant DNA replication initiation at a single origin [39]. The second feature was pronounced accumulation of SNPs around SSRs. Here, loss of the HR factors has somewhat different effects: for *RAD51* KO, a decrease in SSR-proximal SNP accumulation was seen, whereas an increase was seen upon *RAD51-3* KO and no change was found after KO of *RAD51-4*. These data suggest that the SNPs reflect the effects of differing roles of the proteins on damage repair. Given that both RAD51 and RAD51-3 loss affects DNA synthesis, while loss of RAD51-4 does not, an explanation could be that the SNPs are generated due to clashes between the transcription and replication machineries at SSRs, providing a source for such pronounced mutation rates at these sites. In *T. brucei* it is known that the Origin Recognition Complex (ORC) binds to potentially all SSRs and its loss affects levels of RNA at these loci [54]. In addition, RNA-DNA hybrids, which have been mapped to sites of replication-transcription clashes in other eukaryotes [64], form prominently at transcription start sites in *T. brucei* [65]. Thus, it seems conceivable that SSRs are also sites of such interaction in *Leishmania*, though no equivalent mapping of ORC or RNA-DNA hybrids has been reported.

Loss of IdU uptake after induced KO of *RAD51* or *RAD51-3* indicates a role for both HR factors in DNA synthesis and DNA replication, but several lines of evidence suggest these roles are not the same: distinct cell cycle timing of γH2A accumulation, distinct patterns of SNP accumulation around SSRs, and differing changes in DNA replication dynamics after their loss. Impaired nucleotide uptake after RAD51 loss appears to be explained by a shift in the programme of *L. major* DNA replication, with loss of efficient DNA replication initiation at the single primary origin in each chromosome and increased subtelomeric DNA replication initiation. The finding that RAD51 KO cells are impaired in growth and nucleotide uptake argues that increased replication from the subtelomeres is insufficient to compensate for loss of the primary initiation events. In the case of RAD51-3, the very modest loss of replication at the main origin, and no clear increase in subtelomeric replication, seems most readily explained by a widespread impairment in genome replication, perhaps because the RAD51 paralogue is needed to guide processes involved in promoting replication in the face of genome-wide impediments. In this regard, the replication phenotypes after loss of RAD51-3 may be comparable with effects seen after mutation of *T. brucei* MCM-BP [66], a poorly understood factor that modulates activity of the replicative MCM helicase [67, 68]. In addition, such a role for *L. major* RAD51-3 may be akin to roles for mammalian Rad51 paralogues on stalled DNA replication forks, with the differing effects of loss of *L. major* RAD51-4 or RAD51-6 suggesting similar functional compartmentalisation amongst the paralogues [13].

How might RAD51 provide a central role in *Leishmania* DNA replication? One possibility is that DNA replication initiation at the single SSR-localised origin in each chromosome is directly driven by RAD51, perhaps even by catalysing HR. Such an origin-specific function for RAD51 or RAD51-directed HR has no precedent and, indeed, would be distinct from origin activity in *T. brucei*, where these sites are conventionally defined by binding ORC [54, 69]. Nonetheless, ORC binding has not been mapped in any *Leishmania* genome and precedents for recombination-directed replication initiation exist in viruses [70–72], bacteria [17],

polypoid archaea [18] and in *Tetrahymena* [73], albeit normally without a focus on defined genome sites. In addition, human Rad51 acts with MCM8-9, an alternate MCM helicase complex, to initiate DNA replication [74]; though this DNA synthesis pathway is origin-independent in humans, no work has tested MCM8 or MCM9 function in any trypanosomatid. Therefore, it remains to be tested if RAD51 associates with DNA replication initiation complexes and/or with the origins of replication.

Alternatively, and perhaps more likely, RAD51 may play a more indirect role in *Leishmania* genome replication. The earliest acting origin-active SSRs in *T. brucei* co-localize with centromeres [54, 75], and recent work has mapped one putative component of the centromere-binding kinetochore, KKT1, to each MFAseq-mapped origin SSR in *L. major* [76]. These loci might then be vulnerable to breakage, such as during chromosome segregation, and RAD51-directed HR may be required to repair any breaks, such as after mitosis in order to allow proper licensing and firing of origins at these loci. Such a scenario could be compatible with data from other eukaryotes of important roles for RAD51-directed HR is maintaining centromere function [77–79]. In *T. brucei*, the presence of further ORC-defined origins in each chromosome may compensate for loss of RAD51 causing impaired centromere-focused origin function, but such a mutation in *Leishmania* might be more detrimental, given the presence of only a single major centromere-focused origin in each chromosome [39].

A distinct suggestion for an indirect role of RAD51 is that the recombinase does not play an origin- or centromere-focused role, but instead is needed to support DNA replication genome-wide, given previous suggestions that replication from a single major origin in each *Leishmania* chromosome would be insufficient to replicate all chromosomes in S-phase [39, 41]. Such a function could be compatible with origin-independent roles of HR-directed replication (discussed above) and, in the absence of RAD51, complete chromosome replication is lost during S-phase, leading to impaired mitosis and reduced numbers of cells that license the main origins. Alternatively, loss of RAD51 may slow progress of replication forks emanating from the single central origin in each chromosome, preventing completion of chromosome duplication in S phase. Either scenario may explain the increased levels of MFA-seq reads in the subtelomeres. Increased subtelomeric MFA-seq signal in *RAD51* KO cells may reflect greater numbers of cells stalling in late- or post-S phase. Alternatively, subtelomeric DNA synthesis may result from the activation of dormant origins in these regions, serving as back-up for the primary SSR-focused DNA replication initiation reaction; thus, when the primary reaction is compromised by loss of RAD51, subtelomere initiation assumes greater importance. As it stands, we do not know the nature of the subtelomeric DNA synthesis, but these data indicate it is distinct from DNA replication emanating from the main SSR-focused origins and is RAD51-independent. Whether or not this subtelomeric DNA replication relates to a recently proposed form of telomere maintenance [34] is worthy of further investigation.

## Methods

### Parasite culture

Cell lines were derived from *Leishmania major* strain LT252 (MHOM/IR/1983/IR). Promastigotes were cultured at 26˚C in M199 or HOMEN medium supplemented with 10% heat-inactivated fetal bovine serum. Transfections were performed with exponentially growing cells with Amaxa Nucleofactor II, using the pre-set program X-001. Transfectants were selected by limiting dilution in 96-well plates in the presence of appropriate antibiotic. DiCre-expressing cells were selected with 10 $\mu$g.mL$^{-1}$ blasticidin. DiCre/Cas9/T7-expressing cells were selected with 10 $\mu$g.mL$^{-1}$ blasticidin and 20 $\mu$g.mL$^{-1}$ hygromycin. Cells expressing the gene of interest

(GOI) flanked by LoxP sites (GOI$^{Flox}$ cells) were selected with 10 μg.mL$^{-1}$ blasticidin, 20 μg.mL$^{-1}$ hygromycin and μg.mL$^{-1}$ puromycin.

## DNA constructs and cell line generation

A background cell line was established in which DiCre is expressed from the ribosomal locus, while both Cas9 and T7 polymerase are expressed from the *tubulin* array. For this, WT cells were transfected with plasmid pGL2339 [80], previously digested with *Pac*I and *Pme*I, to generate DiCre-expressing cells. Correct integration into the ribosomal locus was confirmed by PCR. Then, the DiCre-expressing cells were transfected with plasmid pTB007 [81], previously digested with *Pac*I, to generate the DiCre/Cas9/T7-expressing cell line. Correct integration of Cas9/T7-encoding cassette was confirmed by PCR. In this way, the Cas9/T7 system, as previously described [81], was used to flank all copies of a GOI with LoxP sites, in a single round of transfection to generate the GOI$^{Flox}$ cell lines used here. Deletion of the GOI was induced by rapamycin-mediated DiCre activation, as previously reported [82, 83].

Donor fragments for Cas9-mediated genome editing were generated by PCR (S1 Table). For this, the ORFs encoding each GOI were PCR-amplified using genomic DNA as template. PCR products of *RAD51* (LmjF.28.0550), *RAD51-4* (LmjF.11.0230), *RAD51-6* (LmjF.29.0450) and *PIF6* (LmjF.21.1190) were cloned between *Nde*I and *Spe*I restriction sites in the vector pGL2314 [83]. PCR product of *RAD51-3* (LmjF.33.2490) was cloned into the *Spe*I restriction site of vector pGL2314. The resulting constructs contained the GOI flanked by LoxP sites (pGL2314*GOI*$^{Flox}$) and were used as templates in PCR reactions to generate the donor fragments flanked by sequences homologues (30 nucleotides) to the targeting integration sites. PCR products were ethanol precipitated and transfected into the DiCre/Cas9/T7-expressing cell line. Correct integration into the expected locus was confirmed by PCR analysis.

The strategy used to generate sgRNAs was essentially as previously described[81]. Briefly, sgRNAs were generated *in vivo* upon transfection with appropriate DNA fragment generated by PCR. These fragments contained the sequence for the T7 polymerase promoter, followed by the 20 nucleotides of sgRNA targeting site and 60 nucleotides of sgRNA scaffold sequence. The Eukaryotic Pathogen CRISPR guide RNA/DNA Design Tool (http://grna.ctegd.uga.edu) was used to generate the 20 nucleotide sequences for sgRNA targeting sites. The default parameters and the highest scoring 20 nucleotide sgRNA sequences were chosen.

## Western blotting

Whole cell extracts were prepared by collecting cells and boiling them in NuPAGE LDS Sample Buffer (ThermoFisher). Extracts were resolved on 4–12% gradient Bis-Tris Protein Gels (ThermoFisher) and then transferred to Polyvinylidene difluoride (PVDF) membranes (GE Life Sciences). Before probing for specific proteins, membranes were blocked with 10% (w/v) non-fat dry milk in phosphate-buffered saline supplemented with 0.05% Tween-20 (PBS-T). Primary antibody incubation was performed for 2 h at room temperature with PBS-T supplemented with 5% non-fat dry milk. Membranes were washed with PBS-T and then incubated with secondary antibodies in the same conditions as the primary antibodies. For HRP-conjugated secondary antibodies, ECL Prime Western Blotting Detection Reagent (GE Life Sciences) was used for band detection as visualized with Hyperfilm ECL (GE Life Sciences). For IRDye-conjugates secondary antibodies, Odyssey Imaging Systems (Li-COR Biosciences) was used for band detection and visualization.

## Antibodies

Generation of affinity-purified antibodies against γH2A (1:1000) from rabbit serum was previously described [84]. Commercial primary antibodies used here were: mouse anti-HA (1: 5000, Sigma), anti-EF1α (1: 40 000, Merck Millipore) and anti-BrdU clone B44 (1: 500, BD Bioscience). Commercial secondary antibodies used here were: goat anti-Rabbit IgG HRP-conjugated (ThermoFisher), goat anti-Rabbit IgG HRP-conjugated (ThermoFisher), goat anti-Rabbit IgG Alexa Fluor 488-conjugated (ThermoFisher), goat anti-Rabbit IgG IRDye 800CW-conjugated (Li-COR Biosciences) and goat anti-Mouse IgG IRDye 680CW-conjugated (Li-COR Biosciences).

## Genome sequencing and bioinformatics analysis

Total DNA extraction was performed with a DNeasy Blood & Tissue Kit (QIAGEN) following the manufacturer's instructions. Genomic DNA libraries were prepared using a Nextera NGS Library Preparation Kit (QIAGEN). Libraries were sequenced at Glasgow Polyomics (www.polyomics.gla.ac.uk/index.html), using a NextSeq 500 Illumina platform, generating paired end reads of 75 nucleotides. Processing of sequencing data was performed at the Galaxy web platform (*usegalaxy.org)[85]*. FastQC (http://www.bioinformatics.babraham.ac.uk/projects/fastqc/) and trimomatic [86] were used for quality control and adapter sequence removal, respectively. BWA-mem[87] was used to map processed reads to the reference genome (*Leishmania major Friedlin* v39, available at Tritrypdb - http://tritrypdb.org/tritrypdb/). Reads with mapping quality score < 30 were discarded using SAMtools [88]. Single nucleotide polymorphisms (SNPs) and InDels were detected using GATK [89] and freebayes [90]. Only those SNPs and InDels with DP of at least 5 and map quality 30 were considered. VCFtools was used to calculate SNP and InDel density, with SNPdensity function[91]. Heatmaps, violin plots and metaplots were generated using Prism Graphpad. Underlying data for metaplots and coverage tracks were generated using deepTools[92]. Mutational SNP signature analysis was performed as previously described [93].

## Marker Frequency Analysis (MFAseq)

After processing, reads were compared essentially using methods described previously [39], though with modifications. Briefly, the number of reads in 0.5 kb windows along chromosomes was determined. The number of reads in each bin was then used to calculate the ratio between exponentially growing and stationary phase cells, scaled for the total size of the read library. Ratio values were converted into Z scores values in a 5 kb sliding window (steps of 500bp), for each individual chromosome. MFAseq profiles for each chromosome were represented in a graphical form using Gviz[94].

## Detection of cells in S phase

Exponentially growing cells were incubated for 30 min with 150 μM IdU. Cells were collected by centrifugation and washed with 1x PBS. Fixation was performed at -20 °C with a mixture (7:3) of ethanol and 1x PBS for at least 16 h. Then, cells were collected by centrifugation and rinsed with washing buffer (1x PBS supplemented with 1% BSA). DNA denaturation was performed for 30 minutes with 2N HCL and reaction was neutralized with phosphate buffer (0.2 M $Na_2HPO_4$, 0.2 M $KH_2PO_4$, pH 7.4). Cells were collected by centrifugation, further incubated in phosphate buffer for 30 mins at room temperature and centrifuged again. To detect incorporated IdU, cells were incubated for 1h at room temperature with anti-BrdU antibody (diluted in washing buffer supplemented with 0.2% Tween-20), collected by centrifugation

and washed with washing buffer. Cells were incubated for 1 h at room temperature with anti-mouse secondary antibody conjugated with Alexa Fluor 488 (diluted in washing buffer supplemented with 0.2% Tween-20), collected by centrifugation and washed with washing buffer. Finally, cells were re-suspended in 1xPBS supplemented with 10 μg.mL⁻¹ Propidium Iodide and 10 μg.mL⁻¹ RNAse A and passed through a 35 μm nylon mesh. Data was acquired with FACSCelesta (BD Biosciences) and further analysed with FlowJo software. Negative control cells, in which anti-BrdU antibody was omitted during IdU detection step, were included in each experiment. Negative control cells were used to draw gates to discriminate positive and negative events.

## Supporting information

**S1 Table. Primers used in this study.**
(TIFF)

**S1 Fig. Combining CRISPR/Cas9 and DiCre to rapidly generate cell lines for inducible knockout.** (A) Cas9 was used to replace all copies of a gene of interest (*GOI*) by a version of the same *GOI* flanked by *LoxP* sites (*GOI^Flox^*); (B) PCR analysis of genomic DNA extracted from the indicated cell lines; approximated annealing positions for primers *a-n* and *b-n* (where n varies from 1 to 5, indicating a distinct sequence for the targeted *GOI* in each cell line) are shown in (A).
(TIFF)

**S2 Fig. Growth profile analysis.** Representative growth curves of the indicated cell lines (red lines) compared to the parental cell line expressing Cas9 and DiCre (black line); growth curves were started with 2 x1 0⁵ cells/ml; cell density was assessed every 24 h (1 day) and error bars depict standard error of the mean (S.E.M.).
(TIFF)

**S3 Fig. Analysis of RAD51 C-terminally tagged with HA.** Cas9 was used to replace all copies of *RAD51* by *RAD51-HA^Flox^*. **(A)** PCR analysis of genomic DNA from the *RAD51-HA^Flox^* cell line; approximate annealing positions for primers *a-1* and *b-1* are as shown in S1A Fig. **(B)** Western blotting analysis of whole cell extracts from *RAD51-HA^Flox^* cell line after 48 h growth without addition (-RAP) or after addition (+RAP) of rapamycin; extracts were probed with anti-HA antiserum and anti-EF1α was used as loading control. **(C)** Representative growth curves of *RAD51-HA^Flox^* cells (red lines) compared to the parental cell line expressing Cas9 and DiCre (black line) in both HOMEM and M199 medium; growth curves were started with 1 x 10⁵ cells/ml; cell density was assessed at the indicated days and error bars depict standard error of the mean (S.E.M.).
(TIFF)

**S4 Fig. Dynamics of KO induction. (A)** Illustration of KO induction scheme; cells were seeded in medium with (+RAP) or without (-RAP) rapamycin; after 4 days (~96 h) of cultivation, cells were re-seeded, cultivated further and then diluted again; all the experiments reported here were performed in cells subjected to this induction protocol; times points indicated in the main figures refer to the second passage (P2, highlighted).**(B)** Illustration of GOI-^Flox^ excision catalyzed by DiCre, as induced by rapamycin. **(C)-(G)** PCR analysis of genomic DNA from the indicated cell lines throughout the indicated passages; DNA was extracted from cells ~72 h of each passage; approximate annealing positions for primers *c* and *d* are shown in (A); (\*) and (\*\*): *GOI^Flox^* and *GOI^Flox^* after excision, respectively.
(TIFF)

**S5 Fig. Analysis of DNA content profile upon prolonged cultivation after KO induction of homologous recombination factors.** Representative histograms from FACS analysis to determine the distribution of cell populations according to DNA content in cells kept in culture for more than 15 passages; 30,000 cells were analysed per sample; 1C and 2C indicate single DNA content (G1) and double DNA content (G2/M), respectively.
(TIFF)

**S6 Fig. Cell cycle progression analysis after replication stress upon KO induction of homologous recombination factors.** The indicated cell lines were left untreated (N.T.) or treated for 8 h with 5 mM HU and then re-seeded in HU-free medium; cells were collected at the indicated time points after HU removal, fixed, stained with Propidium Iodide, and analysed by FACS; 1C and 2C indicate single DNA content (G1) and double DNA content (G2/M), respectively.
(TIFF)

**S7 Fig. Whole genome analysis of InDel accumulation patterns upon KO induction of single or combined homologous recombination factors. (A)** $GOI^{Flox}$ cell lines were grown in the absence (-) or presence (+) of Rapamycin (RAP). Genomic DNA was extracted at P2 and P6 and subjected to deep sequencing. **(B)** InDels relative to the reference genome were identified. Events common to P2 and P6, with or without RAP, were discarded. Events exclusively found in P2 or P6 were considered for the following analysis. **(C)** Quantification of the number of new InDels detected inP2 and P6; data are represented as violin plots, where shape indicates the distribution of pooled data and horizontal dotted white lines indicate the median; differences were tested with Mann-Whitney test; * P<0.05, **P<0.005 and ***P<0.001 **(D)** Heatmaps representing density of new InDels (InDels/Kb) detected in the indicated passages; numbers at the top of each row indicate Pearson correlation between InDel density and chromosome size; when correlation is significant, it is indicated by * P<0.05, **P<0.005 and ***P<0.001. **(E)** Metaplots of normalized density of InDels (InDels/Kb) in passages P2 and P6is plotted +/- 30 Kb around the centre of either $SSR^{ORI}$ ($n = 36$) or $SSR^{non-ORI}$ ($n = 95$) for the indicated cell lines.
(TIFF)

**S8 Fig. SNP mutation signature upon KO of RAD51 related genes.** SNPs were ordered by class (C>A/G>T, C>G/G>C, C>T/G>A, T>A/A>T, T>C/A>G, T>G/A >C) and subsequently subclassified according to immediate flanking sequence: 5′ base (A, C, G, T) before 3′ base (A, C, G, T).
(TIFF)

**S9 Fig. Genotoxic stress resistance profiles upon KO induction of RAD51 and RAD51-3. (A)** Experimental design to evaluate resistance to genotoxic agents as shown in (B-D); cells were seeded in medium with (+RAP) or without (-RAP) rapamycin, in the absence of any genotoxic drug; after 96 h of growth, cells were re-seeded in medium with or without genotoxic agents at various concentrations; after 96 h growth (P2), cell density in each condition was determined. **(B–D)** Relative growth of cells incubated with the indicated concentration of the indicated genotoxic agents, during P2; growth in each concentration is expressed as a percentage of proliferation relative to cells cultivated without the genotoxic drugs; error bars depict SD.
(TIFF)

**S10 Fig. Whole genome analysis of SNPs and InDel accumulation patterns after replication stress upon KO induction of RAD51 and RAD51-3. (A)** $GOI^{Flox}$ cell lines were grown

in the absence (-) or presence (+) of Rapamycin (RAP) for 2 passages (P2). Genomic DNA was extracted from exponentially (*Exp*) growing cells in P2. Then, cells were incubated with 5 mM HU for 8 h. Cells were re-seeded in HU-free medium and after (*After*) 96 h genomic DNA extracted and subjected to deep sequencing. **(B)** SNPs and InDels relative to the reference genome were identified. Events common to *Exp* and *After* cells were discarded. Only events exclusively found in *Exp* or *After* were considered for the following analysis. **(C)** Quantification of the number of new InDels detected in passages *Exp* and *After*. Data are represented as wviolin plots, where shape indicates the distribution of pooled data and horizontal dotted white lines indicate the median; differences were tested with Mann-Whitney test; * $P<0.05$, **$P<0.005$ and ***$P<0.001$. **(D)** Heatmaps representing density of SNPs (SNPs/Kb) and InDels (InDels/Kb) detected in *Exp* and *After*; numbers at the top of each row indicate Pearson correlation between SNPs density or InDels density and chromosome size; when correlation is significant, it is indicated by * $P<0.05$, **$P<0.005$ and ***$P<0.001$. **(E)** Metaplots of normalized density of new SNPs (SNPs/Kb) and new InDels (InDels/Kb), respectively, detected 96 h after cells were released from HU treatment are plotted +/- 30 Kb around the centre of either $SSR^{ORI}$ ($n = 36$) or $SSR^{non-ORI}$ ($n = 95$) for the indicated cell lines.
(TIFF)

**S11 Fig. SNP mutation signature upon replication stress after KO induction of RAD51 and RAD513.** SNPs were ordered by class (C>A/G>T, C>G/G>C, C>T/G>A, T>A/A>T, T>C/A>G, T>G/A >C) and subsequently subclassified according to immediate flanking sequence: 5′ base (A, C, G, T) before 3′ base (A, C, G, T).
(TIFF)

**S12 Fig. Combining CRISPR/Cas9 and DiCre to rapidly generate double KO cell lines. (A)** *i)* Cas9 was used to replace all copies of a gene of interest (*GOI-1*) by a version of the same *GOI* flanked by *LoxP* sites (*GOI-1^Flox^*); *ii)* in the same cell line, Cas9 was used to replace all copies of another gene of interest (*GOI-2)* by a Neomycin resistance gene (NEO); **(B)** PCR analysis of genomic DNA extracted from the indicated cell lines; approximate annealing positions for primers *a-n* and *b-n* (where n varies from 1 to 5, indicating a distinct sequence for the targeted *GOI* in each cell line) are shown in (A).
(TIFF)

**S13 Fig. Dynamics of KO induction.** Induction was performed as depicted in S3A Fig. **(A)** Illustration of GOI^Flox^ excision catalyzed by DiCre, as induced by rapamycin, to generate double KO cells. **(B)—(D)** PCR analysis of genomic DNA from the indicated cell lines throughout the indicated passages; approximate annealing positions for primers *c* and *d* are shown in (A); (*) and (**), *GOI^Flox^* and *GOI^Flox^* after excision, respectively.
(TIFF)

**S14 Fig. Effects of double KO of RAD51 paralogues. (A)** Representative growth curves of the indicated cell lines in the presence or absence of RAP; cells were seeded at $10^5$ cells/ml in day 0 and re-seeded every 4–5 days to complete five passages (P1 to P5); growth profile was also evaluated after cells were kept in culture for more than 15 P (>P15); cell density was assessed every 24 h and error bars depict standard error of the mean (S.E.M.). **(B)** Representative histograms from FACS analysis to determine the distribution of cell population according to DNA content in cells kept in culture for more than 15 P; 30,000 cells were analysed per sample; 1C and 2C indicate one DNA content (G1) and double DNA content (G2/M), respectively.
(TIFF)

**S15 Fig. Whole genome analysis of InDels accumulation patterns upon double KO of RAD51-3 and RAD51-4. (A)** Quantification of the number of new InDels detected after P6 and P2; data are represented as violin plots, where shape indicates the distribution of pooled data and horizontal doted white lines indicate the median. **(B)** Heatmaps representing density of new InDels (InDels/Kb) detected after 4 P in each chromosome. **(C)** Metaplots of normalized density of new InDels (InDels/Kb) after 4P are plotted +/- 30 Kb around the centre of either $SSR^{ORI}$ ($n = 36$) or $SSR^{non\text{-}ORI}$ ($n = 95$) for the indicated cell lines.
(TIFF)

**S16 Fig. Genome-wide mapping of replication initiation upon RAD51 and RAD51-3 KO and HU treatment.** Graphs show the distribution of sites of DNA synthesis initiation across the indicated chromosomes in the indicated cell lines, in each case grown in the absence (-RAP) or the presence (+RAP) of rapamycin. MFA-seq profiles are also shown for cells after incubation with 5 mM HU for 8 h. MFA-seq is represented by Z-scores across the chromosomes, calculated by comparing read depth coverage of DNA from exponentially growing cells relative to stationary cells; the bottom track for each chromosome displays coding sequences, with genes transcribed from right to left in red, and from left to right in blue.
(TIFF)

## Acknowledgments

We thank all current and previous members of the McCulloch and Tosi labs for input, and S. Duncan and J.Mottram for discussions regarding CRISPR/Cas9 and DiCre.

## Author Contributions

**Conceptualization:** Jeziel D. Damasceno, Richard McCulloch.

**Data curation:** Kathryn Crouch, Daniella Bartholomeu.

**Formal analysis:** João Reis-Cunha, Dario Beraldi, Craig Lapsley.

**Funding acquisition:** Luiz R. O. Tosi, Richard McCulloch.

**Investigation:** Jeziel D. Damasceno, João Reis-Cunha, Craig Lapsley.

**Methodology:** Jeziel D. Damasceno, João Reis-Cunha, Kathryn Crouch, Dario Beraldi, Luiz R. O. Tosi, Daniella Bartholomeu, Richard McCulloch.

**Project administration:** Luiz R. O. Tosi, Richard McCulloch.

**Resources:** Kathryn Crouch, Luiz R. O. Tosi, Daniella Bartholomeu, Richard McCulloch.

**Software:** Jeziel D. Damasceno, João Reis-Cunha, Kathryn Crouch, Dario Beraldi, Daniella Bartholomeu.

**Supervision:** Jeziel D. Damasceno, Luiz R. O. Tosi, Daniella Bartholomeu, Richard McCulloch.

**Validation:** Jeziel D. Damasceno, João Reis-Cunha, Craig Lapsley.

**Visualization:** Jeziel D. Damasceno, João Reis-Cunha, Kathryn Crouch, Dario Beraldi.

**Writing – original draft:** Jeziel D. Damasceno, Richard McCulloch.

**Writing – review & editing:** Jeziel D. Damasceno, João Reis-Cunha, Kathryn Crouch, Dario Beraldi, Craig Lapsley, Luiz R. O. Tosi, Daniella Bartholomeu, Richard McCulloch.

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
