## [Decision Letter · Decision Letter 0]

4 Jan 2020

Dear Dr McCulloch,

Thank you very much for submitting your Research Article entitled 'Conditional knockout of RAD51-related genes in Leishmania major reveals a critical role for homologous recombination during genome replication' to PLOS Genetics. Your manuscript was fully evaluated at the editorial level and by independent peer reviewers. The reviewers appreciated the attention to an important problem, but raised some substantial concerns about the current manuscript. Based on the reviews, we will not be able to accept this version of the manuscript, but we would be willing to review again a much-revised version. We cannot, of course, promise publication at that time.

If you decide to revise the manuscript for further consideration at PLOS Genetics, please aim to resubmit within the next 60 days, unless it will take extra time to address the concerns of the reviewers, in which case we would appreciate an expected resubmission date by email to plosgenetics@plos.org.

[LINK]

We are sorry that we cannot be more positive about your manuscript at this stage. Please do not hesitate to contact us if you have any concerns or questions.

Yours sincerely,

Marc Ouellette

Guest Editor

PLOS Genetics

Gregory Barsh

Editor-in-Chief

PLOS Genetics

Dear Richard,

Your article has been reviewed by three experts in the field. They acknowledged the quantity and quality of the work submitted. However the three were quite critical about the solidity of the link between RAD51 and DNA replication. I believe that additional work and explanations would be helpful to establish this link further. There are a number of other specific critiques listed by the three reviewers that you would need to be addressed.

Reviewer's Responses to Questions

**Comments to the Authors:**

Reviewer #1: PGENETICS-D-19-01903

In this paper, Damasceno and colleagues scrutinized the biological consequences of conditional ablation of RAD51 and three RAD51-related proteins in Leishmania major. Loss of RAD51 or RAD51-3, but not RAD51-4 or RAD51-6, impairs DNA synthesis by affecting the initiation of DNA replication at the main origins. Interestingly, ablation of RAD51, RAD51-3, and RAD51-4 genes results in genome-wide mutagenesis. The authors also report that RAD51 plays an unexpected role for proper activation of DNA replication origins. The conclusions are drawn using several complementary approaches. The tools and conditional gene knockouts described here will be useful for the Leishmania major community.

Specific comments:

1. I have difficulty to reconcile the growth curves (Fig. 1E) to the DNA replication defects observed in Fig. 2. Why there is no apparent growth inhibition at P2, while DNA synthesis is impaired for RAD51 and RAD51-3 ? This does not make sense to me. Furthermore, the accumulation of DSBs (Fig. 2C) should also lead to growth defects.

2. The authors should explain why they have induction of DSBs without RAP especially at 0h (RAD51) and 3-6h ?

3. Page 6. “KO induction of RAD51 or RAD51-3 resulted in a pronounced increase of �H2A levels in cells navigating through S-phase up to G2/M”. The data supporting this claim is missing unless they have FACS analysis of synchronized cultures.

4. Page 6. “L. major cell cycle checkpoints do not seem to be enacted by the gene KOs”. This should be shown experimentally, as this is quite an important point, which would distinguish L. major from higher eukaryotic cells.

5. The genome-wide mutagenesis analysis is very interesting. I think this should be better defined in terms of DNA signatures. For instance, in breast cancer, an analysis of 560 whole genomes showed that ‘Signature 3’ corresponds to a deficiency in the HR machinery (see Nature 534, 47–54 (2016) and Nat. Med. 23, 517–525 (2017)). Could the authors better define the mutagenesis signature observed after Rad51, Rad51-3, Rad51-4 depletion ? In addition, why RAD51-6 was not included as a control ?

6. Rad51 is known to physically interact with the replicative MCM helicase in both humans and yeast (Biochem. Biophys. Res. Commun. 2005, 329, 1240–1245; Mol. Cell. Biol. 2008, 28, 1724–1738). It would be important to show that there is a direct interaction with RAD51 and DNA replication components by using immunoprecipitation from their HA-tagged Rad51 strain in hand. This would reveal whether the link between RAD51 and the DNA replication machinery is direct or indirect.

7. In the same line of thoughts it is shown that loss of RAD51 leads to a more pronounced decrease in replication initiation activity at the main replication origins. Can the authors ChIP RAD51 at these origins ?

Minor comments

Fig. 2B. All panels. IdU Fluorescnec should read Fluorescence.

Reviewer #2: Please see attached file with comments.

Reviewer #3: Manuscript shows an analysis of null mutants lacking RAD51 and paralogous concerning DNA replication. The first conclusion that can be taken by results is that null mutantants of the RAD51 paralogues RAD51-4 and RAD51-6 can be generated in L. major. Another conclusion is that conditional gene KO loss of RAD51, RAD51-3 and RAD51-4 led to genome-wide increases in SNPs, demonstrating a widespread role for these related proteins in Leishmania genome stability. I agree with authors about these conclusions. Then, the manuscript discusses the role of RAD51 paralogous onto DNA replication. Although data are very well presented I cannot understand the interpretation of some data proposed by Damasceno et al.

Introduction: Damasceno et al say that “origin number and distribution in Leishmania is unusual, since one study detected only a single site of DNA replication initiation per chromosome [39], while a later study suggested >5000 sites [40]. These data indicate either a pronounced paucity or huge overabundance of origins relative to all other eukaryotes. Alternatively, the disparity in the datasets may be due to a widespread, unconventional route for initiation of DNA synthesis, acting alongside a small number of conventional origins, perhaps indicating novel strategies for DNA replication that may link with genome plasticity.”

This proposed alternative is based on the premise that MFA-seq (methodology used in ref 39) would detect conventional origins while SNS-seq (methodology used in ref 40) would detect an unconventional initiation. I can not see the rational to explain these differences.

Results:

Figure 3C: you say that “the smaller chromosomes accumulated a higher density of new SNPs than the larger chromosomes”. It is difficult to see it in the figure. A graph of correlation containing chromosome size X SNP together with the correlation coefficient (r value) is more indicated to allow this kind of conclusion. Still concerning this figure, authors say that “If not gene expression, then the chromosome size-dependence of SNP density might reflect the limitations of predominant DNA replication initiation at a single origin”. If there is a sinlge origin in every chromosome, why short chromosomes would present more SNP?

Figure S10: * is not explained in the legend. Actually, the entire figure is complicated and difficult to understand.

Figure S13 is lacking.

Since both RAD51 and RAD51_3 are involved with DNA replication (since both KO affect IdU incorporation), I did not understand why authors did not check the mutant without both proteins. Would be these mutants able to replicate?

How authors explain the role of RAD51_3 in DNA replication since lacking of this protein does not change origin profile?

Discussion:

Lacking of RAD51 causes impaired nucleotide uptake with loss of efficient DNA replication initiation at the single primary origin in each chromosome and increased subtelomeric DNA replication initiation. Therefore authors argue that increased replication from the subtelomeres is insufficient to compensate for loss of the primary initiation events. However, Damasceno et al are not considering that fork speed can be altered in the absence of RAD51. It hs been already been showed that disrupted functions of RAD51 will lead to a reduced replication fork speed (RADIATION DNA DAMAGE AND USE IN CANCER/ THERAPEUTICSTRANSLATION OF RADIATION MODIFIERS E. Dikomey*, K. Borgmann*, S. Köcher*, M. Kriegs*, W. Mansour*, A.C. Parplys*, T. Rieckmann*, **, K. Rothkamm* In: in DNA Repair in Cancer Therapy (Second Edition), 2016)

The discussion that RAD51 could be the initiator in the replication process is very, very speculative. I believe that any propose in this direction should arrive after demonstration that (i) ORC KO replicates well and (ii) RAD51 localizes with MFA-seq origins.

The increment of origins after RAD51 KO could be due the fact that: single main origin conflicts with transcription (or some genomic structure), RAD51 is absent to fix and solve stalled/collapsed forks; then dorment origins are fired. I did not understand why this simplest interpretation was not considered.

**Have all data underlying the figures and results presented in the manuscript been provided?**

Reviewer #1: Yes

Reviewer #2: Yes

Reviewer #3: Yes

PLOS authors have the option to publish the peer review history of their article (what does this mean?). If published, this will include your full peer review and any attached files.

Reviewer #1: Yes: Jean-Yves Masson

Reviewer #2: No

Reviewer #3: No

---

## [Decision Letter · Decision Letter 1]

5 May 2020

Dear Dr McCulloch,

We are pleased to inform you that your manuscript entitled "Conditional knockout of RAD51-related genes in Leishmania major reveals a critical role for homologous recombination during genome replication" has been editorially accepted for publication in PLOS Genetics. Congratulations!

Yours sincerely,

Marc Ouellette

Guest Editor

PLOS Genetics

Gregory Barsh

Editor-in-Chief

PLOS Genetics

Comments from the reviewers (if applicable):

The answers and rebuttal were appropriate for a fine manuscript.

Reviewer's Responses to Questions

**Comments to the Authors:**

Reviewer #1: The authors have satisfactorily responded to all my questions. They have made the necessary changes to the manuscript and assessed the criticisms to be the best of their capabilities. Although some biological effects are still difficult to explain, I feel the manuscript is suitable fo PloS Genetics and represent an important advance for the field.

Reviewer #3: All questions were addressed

**Data Deposition**

http://datadryad.org/submit?journalID=pgenetics&manu=PGENETICS-D-19-01903R1

**Press Queries**

---

## [Editor Report · Acceptance letter]

21 Jun 2020

PGENETICS-D-19-01903R1 

Conditional knockout of RAD51-related genes in Leishmania major reveals a critical role for homologous recombination during genome replication 

Dear Dr McCulloch, 

We are pleased to inform you that your manuscript entitled "Conditional knockout of RAD51-related genes in Leishmania major reveals a critical role for homologous recombination during genome replication" has been formally accepted for publication in PLOS Genetics! Your manuscript is now with our production department and you will be notified of the publication date in due course.

With kind regards,

Kaitlin Butler

PLOS Genetics

On behalf of:
